# A Unified Analysis of Mixed Sample Data Augmentation: A Loss Function Perspective

**Chanwoo Park** [*,†]
MIT
cpark97@mit.edu

**Sangdoo Yun** [*]
NAVER AI Lab
sangdoo.yun@navercorp.com

**Sanghyuk Chun**
NAVER AI Lab
sanghyuk.c@navercorp.com

## Abstract

We propose the first unified theoretical analysis of mixed sample data augmentation (MSDA), such as Mixup and CutMix. Our theoretical results show that regardless of the choice of the mixing strategy, MSDA behaves as a pixel-level regularization of the underlying training loss and a regularization of the first layer parameters. Similarly, our theoretical results support that the MSDA training strategy can improve adversarial robustness and generalization compared to the vanilla training strategy. Using the theoretical results, we provide a high-level understanding of how different design choices of MSDA work differently. For example, we show that the most popular MSDA methods, Mixup and CutMix, behave differently, *e.g.*, CutMix regularizes the input gradients by pixel distances, while Mixup regularizes the input gradients regardless of pixel distances. Our theoretical results also show that the optimal MSDA strategy depends on tasks, datasets, or model parameters. From these observations, we propose generalized MSDAs, a Hybrid version of Mixup and CutMix (**HMix**) and Gaussian Mixup (**GMix**), simple extensions of Mixup and CutMix. Our implementation can leverage the advantages of Mixup and CutMix, while our implementation is very efficient, and the computation cost is almost neglectable as Mixup and CutMix. Our empirical study shows that our HMix and GMix outperform the previous state-of-the-art MSDA methods in CIFAR-100 and ImageNet classification tasks. Source code is available at https://github.com/naver-ai/hmix-gmix.

## 1 Introduction

As deep neural networks (DNNs) are data-hungry, the scale of datasets has become a foundation of modern DNN training; recent ground-breaking deep models are built upon gigantic datasets, such as 410B language tokens [6], 3.5B images [47], and 1.8B image-text pairs [33]. While such tremendously large-scale datasets are not always collectible, amplifying the dataset scale by synthesizing more data points by data augmentation techniques is common. Especially, *mixed sample data augmentation* (MSDA) [76, 63, 62, 30, 72, 66, 60, 3, 68, 74, 11, 65, 39, 25, 18, 54, 32, 16, 23, 38, 45, 14, 32, 58, 44] has become a standard technique to train a strong deep model by synthesizing a mixed sample from multiple (usually two) samples by combining both of their sample values and labels in a linear combination [76] or a cut-and-paste manner [72]. This simple idea, however, shows surprising performance enhancements in various applications, including image object recognition [72, 39, 25, 64, 16], semi-supervised learning [4, 59] self-supervised learning [35, 40, 42], noisy label training [43], meta-learning [71], semantic segmentation [10, 20], natural language understanding [24, 34], and audio processing [48, 37, 49]. Another advantage of MSDA beyond the performance improvements is that MSDA usually does not need domain-specific knowledge, such as strong image-specific [15] or audio-specific [53] transformations; hence MSDA can be universally employed

---

[*]Equal contribution   [†] Works done while doing an internship at NAVER AI Lab

36th Conference on Neural Information Processing Systems (NeurIPS 2022).

by various applications. However, although MSDA shows excellent benefits in practice, there is still yet not enough understanding of how MSDA works well universally; *can MSDA always show better generalization and robustness than the standard training with a theoretical guarantee?* Furthermore, the design choice of MSDA can be significantly varying and the optimal design choice is still ambiguous. For example, Lee *et al*. [42] showed that in self-supervised learning, Mixup is more effective than CutMix, while other studies [56, 57] observed opposite results. The ambiguity is originated from the fact that we do not have a unified theoretical lens of understanding how different design choices affect the actual learning process; in short, *how are Mixup and CutMix different?*

There have been several attempts to theoretically understand Mixup, a special case of MSDA [77, 12, 8, 78]. They delve into the effect of Mixup in a loss function perspective, *e.g*., Mixup behaves as a regularization of the standard training [77], or in a learning theory perspective, *e.g*., Mixup training can provide an upper bound for the true loss [12, 71]. However, their analyses are limited to Mixup, while other MSDAs, such as CutMix, are poorly understandable through the lens of their analyses. In this paper, we extend the theoretical results of Zhang *et al*. [77] and Chidambaram *et al*. [12] to a general MSDA to provide a first unified theoretical lens for understanding how general MSDAs work by different choices of mixing strategies. We show that MSDA behaves as an input gradient and Hessian regularization (Theorem 1) as well as a regularizer for the first layer parameters; MSDA improves adversarial robustness (Theorem 3) and generalization (Theorem 4). Our theoretical results show that popular MSDA methods, such as Mixup and CutMix, behave differently in terms of regularization effects. Briefly, CutMix gives a strong regularization in the product of nearby distance pixel-level partial gradient and nearby distance Hessian of the estimated function $f$, while CutMix gives a weak regularization in the product of long-distance pixel-level partial gradient and long-distance Hessian of the estimated function $f$. In contrast, Mixup gives a regularization in gradient or Hessian of the estimated function $f$ regardless of the pixel-level distance.

From our unified theoretical lens for MSDA, we can conclude that *there is no one-fit-all optimal MSDA fit to every data or model parameter*. In other words, the optimal mixing strategy depends on applications, datasets, and model architectures. It supports previous empirical observations that combining different MSDA methods (*e.g*., alternatively using Mixup and CutMix during training) can outperform using only one MSDA [50, 5, 64, 73]. From these observations, we propose two simple MSDA methods that naturally generalize Mixup and CutMix, so that it can take advantage of both methods. Our first proposed method, Hybrid version of Mixup and CutMix **(HMix)**, mixes two samples in both Mixup and CutMix manners; it first cut-and-paste two samples as CutMix, and then it linearly interpolates the out-of-box values of two samples as Mixup. We let HMix be able to behave as both Mixup and CutMix by introducing a stochastic control parameter. Our second proposed method, Gaussian Mixup **(GMix)**, also mixes two samples in both Mixup and Cutmix manners; firstly we select a point, and then we mix two samples gradually using the Gaussian function. Our empirical results on CIFAR-100 and ImageNet show that HMix and GMix outperform the state-of-the-art MSDA methods, including Mixup, CutMix, and Stochastic Mixup & CutMix.

## 2   A General Framework for Mixed Sample Data Augmentation (MSDA)

In this section, we define the formal definition of MSDA and notations. We define a training dataset as $D = \{z_i = (x_i, y_i)\}_{i=1}^m$, randomly sampled from a distribution $\mathcal{P}_z$. Here, $z = (x, y)$ is the input (*e.g*., an image) and output (*e.g*., the target class label) pair. Then, for randomly selected two data samples, $z_i$ and $z_j$, an augmented sample by MSDA, $\tilde{z}_{i,j}^{(\text{MSDA})}$, is synthesized as follows

$$\tilde{z}_{i,j}^{(\text{MSDA})}(\lambda, 1 - \lambda) = (\tilde{x}_{i,j}^{(\text{MSDA})}(\lambda, 1 - \lambda), \tilde{y}_{i,j}^{(\text{MSDA})}(\lambda, 1 - \lambda))$$

$$\text{where,} \quad \tilde{x}_{i,j}^{(\text{MSDA})}(\lambda, 1 - \lambda) = M(\lambda) \odot x_i + (1 - M(\lambda)) \odot x_j \quad \text{and} \quad (1)$$

$$\tilde{y}_{i,j}^{(\text{MSDA})}(\lambda, 1 - \lambda) = N(\lambda) \odot y_i + (1 - N(\lambda)) \odot y_j,$$

where $\lambda$ is the ratio parameter between samples, drawn from $\mathcal{D}_\lambda$ (usually Beta distribution). $\odot$ means a component-wise multiplication in vector (or matrix). $M(\lambda)$ is a random variable conditioned on $\lambda$ that indicates how we mix the input (*e.g*., by linear interpolation [76] or by a pixel mask [72]). $N(\lambda)$ denotes a random variable conditioned on $\lambda$ that demonstrates how we combine the output. We assume that the output $y$ can be one-dimensional data or a matrix; the former means regression or classification task, and the latter means semantic segmentation task. For the sake of simplicity, we let $y$ be one-dimensional data: $N(\lambda) = \lambda$ and $\mathbb{E}[M(\lambda)] = \lambda\vec{1}$.

**Remark 1.** If the meaning is not ambiguous, then we sometimes omit $\lambda$ (*i.e.*, $M(\lambda)$ to $M$). For the sake of simplicity, we consider mixing only two samples (*i.e.*, $\tilde{z}_{i,j}^{(\text{MSDA})}(\lambda, 1-\lambda)$), but we can similarly extend these analyses to mixing $n$-samples data augmentation [60, 32, 23]. If we combine $n$-samples, the ratio parameter will be a vector in general (See Appendix B).

**Remark 2.** As recent studies [68, 39, 38] have shown, $M(\lambda)$ or $N(\lambda)$ can depend on $(z_i, z_j)$, *e.g.*, by using a saliency map [39, 38] or the class activation map [68]. Since the proof techniques for our theoretical analysis are invariant to the choice of $M(\lambda)$ and $N(\lambda)$, our proof techniques also can be applied to the dynamic MSDA methods. For simplicity, we assume that $M$ is a random variable only depending on $\lambda$. In other words, we assume $\mathcal{W}$ as a random sample space, and $M : \mathcal{W} \times \Lambda \to \mathbb{R}^n$ is a measurable function. We left the theoretical analysis of dynamic methods to the future.

Now, we re-write the two most popular MSDA methods, Mixup [76] and CutMix [72], for $i$-th and $j$-th samples with $\lambda$, drawn from $\mathcal{D}_\lambda$, by using the proposed framework (Equation (1)) as follows:

$$
\begin{aligned}
\textbf{Mixup:} \quad &\tilde{z}_{i,j}^{(\text{mixup})}(\lambda, 1-\lambda) = (\tilde{x}_{i,j}^{(\text{mixup})}(\lambda, 1-\lambda), \tilde{y}_{i,j}^{(\text{mixup})}(\lambda, 1-\lambda)) \\
&\text{where} \quad \tilde{x}_{i,j}^{(\text{mixup})}(\lambda, 1-\lambda) = \lambda x_i + (1-\lambda)x_j \quad \text{and} \\
&\qquad\qquad \tilde{y}_{i,j}^{(\text{mixup})}(\lambda, 1-\lambda) = \lambda y_i + (1-\lambda)y_j.
\end{aligned}
\tag{2}
$$

$$
\begin{aligned}
\textbf{CutMix:} \quad &\tilde{z}_{i,j}^{(\text{cutmix})}(\lambda, 1-\lambda) = (\tilde{x}_{i,j}^{(\text{cutmix})}(M, 1-M), \tilde{y}_{i,j}^{(\text{cutmix})}(\lambda, 1-\lambda)) \\
&\text{where} \quad \tilde{x}_{i,j}^{(\text{cutmix})}(\tilde{M}^{(\text{cutmix})}, 1-\tilde{M}^{(\text{cutmix})}) = \tilde{M}^{(\text{cutmix})} \odot x_i + (1-\tilde{M}^{(\text{cutmix})}) \odot x_j \\
&\text{and} \quad \tilde{y}_{i,j}^{(\text{cutmix})}(\lambda, 1-\lambda) = \lambda y_i + (1-\lambda)y_j.
\end{aligned}
\tag{3}
$$

Note that Equation (2) is equivalent to Equation (1) by putting $M(\lambda) = \lambda \vec{1}$. In Equation (3), $\tilde{M}^{(\text{cutmix})}$ is a binary mask that indicates the location of the cropped box region with a relative area $\lambda$. Similarly, other MSDA variants can be easily formed as Equation (1) by introducing new $M(\lambda)$ and $N(\lambda)$.

**Notations.** We define the loss function as $l(\theta, z)$, where $\theta \in \Theta \subseteq \mathbb{R}^d$. We define $L(\theta) = \mathbb{E}_{z \sim \mathcal{P}_z} l(\theta, z)$ as the non-augmented population loss and $L_m(\theta) = \frac{1}{m} \sum_{i=1}^m l(\theta, z_i)$ as the empirical loss for the non-augmented population. For a general MSDA, we can define MSDA loss as

$$
L_m^{\text{MSDA}}(\theta) = \mathbb{E}_{i,j \sim \text{Unif}([m])} \mathbb{E}_{\lambda \sim \mathcal{D}_\lambda} \mathbb{E}_M l(\theta, \tilde{z}_{i,j}^{(\text{MSDA})}(\lambda, 1-\lambda)).
\tag{4}
$$

Therefore, the Mixup and CutMix losses can be written as

$$
\begin{aligned}
L_m^{\text{mixup}}(\theta) &= \frac{1}{m^2} \sum_{i,j=1}^m \mathbb{E}_{\lambda \sim \mathcal{D}_\lambda} l(\theta, \tilde{z}_{i,j}^{(\text{mixup})}(\lambda, 1-\lambda)) \\
L_m^{\text{cutmix}}(\theta) &= \frac{1}{m^2} \sum_{i,j=1}^m \mathbb{E}_{\lambda \sim \mathcal{D}_\lambda} \mathbb{E}_M l(\theta, \tilde{z}_{i,j}^{(\text{cutmix})}(\lambda, 1-\lambda)),
\end{aligned}
\tag{5}
$$

where $[m] = \{1, 2, \ldots, m\}$ and $\mathcal{D}_\lambda$ is a distribution supported on $[0, 1]$ with a conjugate prior. Throughout this paper, we consider $\mathcal{D}_\lambda$ as $\text{Beta}(\alpha, \beta)$, a common selection for $\lambda$ in practice. We define $\mathcal{D}_X$ as the empirical distribution of the training dataset.

## 3 A Unified Theoretical Understanding of MSDA

In this section, we provide a unified theoretical lens of how MSDA works. Specifically, we follow the theoretical results for Mixup provided by Zhang *et al.* [77], where Zhang *et al.* have shown that Mixup is equivalent to the summation of the original loss function and a Mixup-originated regularization term. We will give a general approximation form for MSDA using $\lambda \sim \text{Beta}(\alpha, \beta)$. We also show that our analysis can be extended to $n$-sample mixed augmentation (See Appendix B)

**From an MSDA loss to an input gradient and Hessian regularization.** We first consider the following class of loss functions for a twice differentiable prediction function $f_\theta(x)$ (*e.g.*, a softmax output of a neural network), a twice differentiable function $h$, and target $y$:

$$
\mathcal{L} = \{l(\theta, z) \mid l(\theta, z) = h(f_\theta(x)) - yf_\theta(x) \text{ for a twice differentiable function } h\}.
$$

This function class $\mathcal{L}$ includes the loss function induced by Generalized Linear Models (GLMs) and cross-entropy. Now, we introduce our first theoretical result that induces an MSDA loss (*i.e.*, Equation (4)) can be re-written as the summation of the original loss (the empirical loss for the non-augmented population loss, $L_m(\theta)$) and input gradient-related regularization terms as follows.

**Theorem 1.** *Consider a loss function $l \in \mathcal{L}$. We define $\tilde{D}_\lambda$ as $\frac{\alpha}{\alpha+\beta}Beta(\alpha+1,\beta)+\frac{\beta}{\alpha+\beta}Beta(\beta+1,\alpha)$. Assume that $\mathbb{E}_{r_x \sim \mathcal{D}_X}[r_x] = 0$. Then, we can re-write the general MSDA loss (4) as*

$$L_m^{MSDA}(\theta) = L_m(\theta) + \sum_{i=1}^{3} \mathcal{R}_i^{(MSDA)}(\theta) + \mathbb{E}_{\lambda \sim \tilde{\mathcal{D}}(\lambda)}\mathbb{E}_M[(1-M)^\intercal\varphi(1-M)(1-M)], \quad (6)$$

*where* $\lim_{a \to 0} \varphi(a) = 0$,

$$\mathcal{R}_1^{(MSDA)}(\theta) = \frac{1}{m}\sum_{i=1}^{m}(y_i - h'(f_\theta(x_i)))\left(\nabla f_\theta(x_i)^\intercal x_i\right)\mathbb{E}_{\lambda \sim \tilde{D}_\lambda}(1-\lambda),$$

$$\mathcal{R}_2^{(MSDA)}(\theta) = \frac{1}{2m}\sum_{i=1}^{m}h''(f_\theta(x_i))\mathbb{E}_{\lambda \sim \tilde{D}_\lambda}\mathbf{G}(\mathcal{D}_X, x_i, f, M), \quad (7)$$

$$\mathcal{R}_3^{(MSDA)}(\theta) = \frac{1}{2m}\sum_{i=1}^{m}(h'(f_\theta(x_i)) - y_i)\mathbb{E}_{\lambda \sim \tilde{D}_\lambda}\mathbf{H}(\mathcal{D}_X, x_i, f, M),$$

*and*

$$\mathbf{G}(\mathcal{D}_X, x_i, f, M) = \mathbb{E}_M(1-M)^\intercal\mathbb{E}_{r_x \sim \mathcal{D}_X}\left(\nabla f(x_i) \odot (r_x - x_i)\left(\nabla f(x_i) \odot (r_x - x_i))^\intercal\right)(1-M)$$
$$= \sum_{j,k \in coord} a_{jk}\partial_j f_\theta(x_i)\partial_k f_\theta(x_i)\left(\mathbb{E}_{r_x \sim \mathcal{D}_X}[r_{xj}r_{xk}] + x_{ij}x_{ik}\right),$$

$$\mathbf{H}(\mathcal{D}_X, x_i, f, M) = \mathbb{E}_{r_x \sim \mathcal{D}_X}\mathbb{E}_M(1-M)^\intercal\left(\nabla^2 f_\theta(x_i) \odot ((r_x - x_i)(r_x - x_i)^\intercal)\right)(1-M)$$
$$= \sum_{j,k \in coord} a_{jk}\left(\mathbb{E}_{r_x \sim \mathcal{D}_X}[r_{xj}r_{xk}\partial_{jk}^2 f_\theta(x_i)] + x_{ij}x_{ik}\partial_{jk}^2 f_\theta(x_i)\right),$$

*where*

$$a_{jk} := \mathbb{E}_M[(1-M_j)(1-M_k)]. \quad (9)$$

*Proof outline of Theorem 1.* Using the definition of $\tilde{z}_{ij}$ and using the fact that the Binomial distribution and Beta distribution are in the conjugate, we can reformulate $L_m^{(MSDA)}$. In the process of reformulating $L_m^{(MSDA)}$, we should define $\tilde{\mathcal{D}}_\lambda$. Then, we can make a quadratic Taylor approximation of the loss term. Here, $\mathbb{E}_{r_x}[r_x] = 0$ is used for not only the simplicity of the results, but also for the fact that using normalization in the dataset. Details can be found in Appendix A. We also show that Theorem 1 can be extended to $n$-sample MSDA methods (Appendix B). In this case, the combinatorial terms in quadratic multivariate Taylor approximation also come out. □

*How is our approximation accurate?* We call $\tilde{L}_m^{MSDA}(\theta) := L_m(\theta) + \sum_{i=1}^{3}\mathcal{R}_i^{(MSDA)}$ as *the approximate MSDA loss*. Here, we empirically demonstrate that our quadratic approximation is almost accurate by following numerical validations in [67, 8, 77]. Specifically, we train logistic regression models on two-moons dataset [7] in two ways: (1) by using the original MSDA loss function (2) by using our approximated loss function. We employ two MSDA examples as below.

- The original Mixup *i.e.*, $\lambda \sim \text{Beta}(1,1)$ and $M = \lambda\vec{1}$
- Variants of CutMix *i.e.*, $\lambda \sim \text{Beta}(1,1)$ and $M = (m_1, m_2)$ such that $m_i \sim \text{Bernoulli}(\lambda)$.

Figure 1 displays the approximate loss function and the original loss function. According to empirical findings, we can conclude that the original MSDA loss is fairly close to the approximate MSDA loss.

*What makes the difference between various MSDA methods?* In the theorem, as we define $E_M(1-M) = 1-\lambda$, $\mathcal{R}_1^{(\text{MSDA})}$ is the same for every MSDA method. Namely, the difference between MSDA methods originated from $\mathcal{R}_2^{(\text{MSDA})}$ and $\mathcal{R}_3^{(\text{MSDA})}$. Note that if we set $M = \lambda\vec{1}$, Theorem 1 indicates a Mixup loss (5), and the result is consistent with Zhang *et al.* [77]. In Equation (7) and Equation (8), we observe that $\mathcal{R}_2$ is related to the input gradient $\nabla f_\theta(x_i)$ and $\mathcal{R}_3$ is related to input

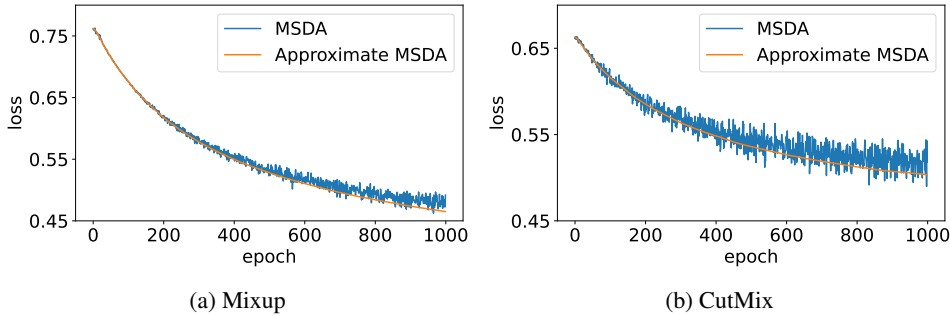

Figure 1: Comparison of the original MSDA loss with the approximate MSDA loss function.

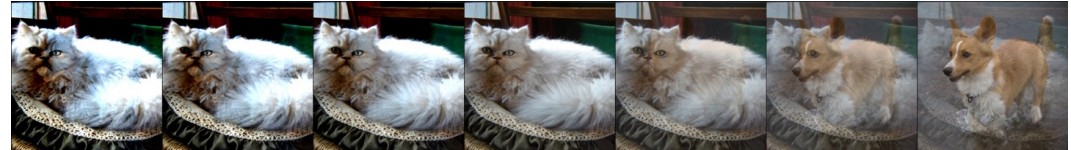

Figure 2: The negative $M$ value results. the image is $\lambda * \mathrm{dog} + (1 - \lambda) * \mathrm{cat}$ where $-0.75 \leq \lambda \leq 0.75$

Hessian $\nabla^2 f_\theta(x_i)$ with mask-dependent coefficients $a_{jk}$ (9). In other words, different design choice of MSDA (*e.g.*, how to design $M$) will lead to different magnitudes of regularization on the input gradients and Hessians. Because the values of input gradients and Hessians are varying by datasets, tasks and model architecture choices, we can conclude that the optimal choice of $M$ is dependent on the applications. We also describe how the other MSDA methods (*e.g.*, dynamic MSDAs [65, 39, 38]) can be interpreted through the lens of our unified analysis in Appendix I.

In addition, to show that MSDA behaves as a regularization on input gradients and Hessians for any desired $a_{jk}$, we also show that there always exists a MSDA design choice $M$ for any desired regularization coefficient matrix $A(\lambda) := (a_{jk}(\lambda))$ with the regular conditions.

**Theorem 2.** *For the given $\lambda$, we assume $A(\lambda) - (1 - \lambda)^2 \vec{1}\vec{1}^{\mathsf{T}}$ is a nonnegative definite matrix. Then we can construct a real-valued mask $M$ that $\mathbb{E}(1 - M_j)(1 - M_k) = a_{jk}$ for all $j, k$.*

*Proof.* Setting $M = 1 - \lambda + (A(\lambda) - (1 - \lambda)^2 \vec{1}\vec{1}^{\mathsf{T}})^{1/2} Z$ where $Z$ is normal distribution, the theorem holds. □

Note that, in the proof, $M$ values are not bounded where typically we choose $0 \leq M_i \leq 1$. In other words, the theorem holds if we allow mask values out of $[0, 1]$. To investigate the potentiality of unbounded mask, we explore Mixup with unbounded masks in Figure 2. Although, allowing negative values to $M$ can be beneficial, we leave a new mask design with unbounded values as a future work.

Unfortunately, as the target loss function (6) is mingled with the choice of mask $M$, data sample $x_i$, and pixel-level function gradient, the optimal choice of mixing strategy $M$ is not achievable in the closed-form solution. Instead, Theorem 1 implies that there is no absolute superiority between the design choice of MSDA, but it depends on datasets and the target tasks, as our empirical observation is consistent with the theoretical interpretation. In Section 4, we will provide more examples of how different $M$ affects the actual coefficients $a_{jk}$ and the input gradients for better understanding.

Using the regularization term $\mathcal{R}_2^{(\mathrm{MSDA})}$ (7), we can also provide a theoretical connection between MSDA methods and the notion of flatness where a more flat solution leads to better generalization in applications [36, 21, 31, 19, 9]. Inspired by Ma *et al.* [46], we split the parameters by $\theta = (\theta_1, \theta_2)$, and then the neural network can be represented by the form $f_\theta(x) = \tilde{f}_{\theta_2}(\theta_1 x)$. Therefore, we have

$$\nabla_{\theta_1} \tilde{f}_{\theta_2}(\theta_1 x) = \frac{\partial f}{\partial(\theta_1 x)} x^{\mathsf{T}}, \qquad \nabla_x \tilde{f}_{\theta_2}(\theta_1 x) = \theta_1^{\mathsf{T}} \frac{\partial f}{\partial(\theta_1 x)},$$

where $\frac{\partial f}{\partial(\theta_1 x)}$ is the partial derivative of the first layer. Now, we have

$$((1-M)\odot\nabla f(x))^\mathsf{T} x = \mathrm{tr}(x((1-M)\odot\nabla f(x))^\mathsf{T}) = \mathrm{tr}\left(x\left(\theta_1^\mathsf{T}\frac{\partial f}{\partial\theta_1 x}\odot(1-M)\right)^\mathsf{T}\right)$$

$$= \mathrm{tr}\left(x\left(\left(\frac{\partial f}{\partial\theta_1 x}\right)^\mathsf{T}\theta_1\,\mathrm{diag}(1-M)\right)\right) = \mathrm{tr}\left(\left(\nabla_{\theta_1}\tilde{f}_{\theta_2}(\theta_1 x)\right)^\mathsf{T}\theta_1\,\mathrm{diag}(1-M)\right).$$

Note that the terms in **G** (8) can be re-written as follows

$$\sum_{j,k\in\mathrm{coord}}\mathbb{E}_M[(1-M_j)(1-M_k)]\partial_j f_\theta(x_i)\partial_k f_\theta(x_i)(x_{ij}x_{ik}) = \mathbb{E}\left[(((1-M)\odot\nabla f(x))^\mathsf{T} x)^2\right].$$

In other words, by minimizing the regularization term $\mathcal{R}_2^{(\text{MSDA})}$, $\int(\theta_1^\mathsf{T}\nabla_{\theta_1}\tilde{f}_{\theta_2})^2$, *i.e.*, the regularization effect of flatness at the interpolation solution can be minimized in a sample-wise weighted manner. Therefore, the regularization term $\mathcal{R}_2^{(\text{MSDA})}$ also can be interpreted as a regularization of the first layer parameters and their partial derivative of $f$.

**Robustness and generalization properties of MSDA.** As a number of studies [51, 52, 46] have shown that regularizing input gradient and Hessian will give better robustness and generalization to the target network $\theta$, it can be shown that MSDA also has adversarial robustness properties and generalization properties based on Theorem 1. The full statement of Theorem 3 and Theorem 4 can be found in Appendix C and Appendix D, respectively.

**Theorem 3** (Informal). *With the logistic loss function under the ReLU network, the approximate loss function of MSDA is greater than the adversarial loss with the $\ell_2$ attack of size $\epsilon\sqrt{d}$.*

*Proof outline of Theorem 3.* Defining adversarial loss function and using second order taylor expansion, we can prove that adversarial loss is less than MSDA loss. $\square$

**Theorem 4** (Informal). *Under the GLM model and the regular conditions, and if we use MSDA in training, we have*

$$L(\theta) \leq \tilde{L}_m^{(\text{MSDA})}(\theta) + \sqrt{\frac{\mathcal{O}\left(\log(1/\delta)\right)}{n}}$$

*with probability at least $1-\delta$. This also holds for the MSE loss and a feature-level MSDA.*

*Proof outline of Theorem 4.* MSDA regularization can be altered to the original empirical risk minimization problem with a constrained function set, and calculating Radamacher complexity of this function set gives the theorem. $\square$

In addition to Theorem 3 and Theorem 4, we can prove that the optimal solution of (4) can achieve a perfect classifier (*i.e.*, classifies every augmented sample $x$ correctly) in the logistic classification setting by following Chidambaram *et al.* [12]. The full statement are in Appendix E.

**Summary.** Our unified theoretical lens for MSDA shows that for any MSDA method formed as Equation (4), the method satisfies that (1) it behaves as a regularizer of input gradients, Hessian, and the first layer parameters (Theorem 1); (2) there exists a mask $M$ for any desired regularization coefficients $a_{jk}$ (Theorem 2) (3) it achieves better adversarial robustness (Theorem 3) and generalization (Theorem 4) than the vanilla training. Interestingly, Theorem 1 shows the difference between different MSDA design choices (*e.g.*, different $M$, such as linear interpolation [76], cropped box [72]) will lead to different magnitudes of the input gradient regularization (7).

# 4  Comparison of Different MSDA Design Choices: The Role of Masks

As we observed in the previous section, different design choices for MSDA (*i.e.*, the choice of $M$) affect to the degree of the regularization in Theorem 1 (*i.e.*, $\mathcal{R}_2^{(\text{MSDA})}$ and $\mathcal{R}_3^{(\text{MSDA})}$) by the relationships of pixels. In this section, we show how different MSDA methods lead to different regularization effects by empirical studies; we first show the values of the regularization coefficients

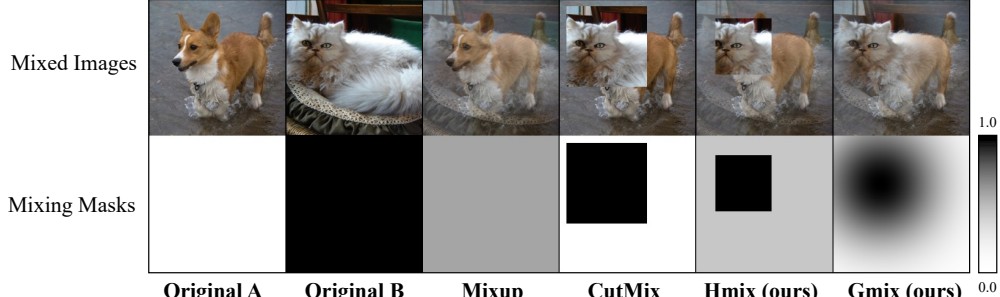

Figure 3: **Examples generated by different MSDAs.** From left to right, two original images to be mixed, Mixup, CutMix sample, HMix, and GMix. The first and the second rows show generated samples and their mixing masks $M$, respectively. We set $\lambda = 0.65$ for all images and $r = 0.5$ for HMix.

$a_{jk}$ by varying masks; then we show the input gradient values that are regularized by $a_{jk}$ (9) after the MSDA training; finally, we show that the best choice of the mask design can be varying by the target task settings. In addition, we propose two generalized versions of Mixup and CutMix, called HMix and GMix, that empirically show the intermediate property of Mixup and CutMix.

**Introduction to HMix and GMix.**  Recall that the regularization coefficients $a_{jk}$ is determined by $M$ (Equation (9)). For example, by choosing $M = \lambda \vec{1}$ (*i.e.*, Mixup), $a_{jk}$ is always $(1 - \lambda)^2$. On the other hand, the result slightly changes for CutMix: $a_{jk}$ depends on how $j$ and $k$ are close. Informally, due to dependency between $M_j$ and $M_k$ (as $M$'s component is always 0 in the cropped box regions and 1 in others), close $j$ and $k$ give large $a_{jk}$, but distant $j$ and $k$ give small $a_{jk}$. $a_{jk}$ is calculated as

$$a_{jk} = \frac{\max(\min(h(j_1) - l(k_1), h(k_1) - l(j_1)), 0) \max(\min(h(j_2) - l(k_2), h(k_2) - l(j_2)), 0)}{(n - [\sqrt{1 - \lambda}n])^2} \quad (10)$$

where $j = (j_1, j_2), k = (k_1, k_2), h(t) = \min(t, n - [\sqrt{1 - \lambda}n]), l(t) = \max(t - [\sqrt{1 - \lambda}n], 0)$.

We visualize $a_{jk}$ of different MSDA methods in Figure 4. We compare Mixup, CutMix, Stochastic Mixup & CutMix. We also propose two generalized MSDA methods, named **HMix** and **GMix**. Before comparing the methods, we first formally define Stochastic Mixup & CutMix, HMix and GMix. These methods can be formed as (1) where the definition of $M$ is varying by the methods.

*Stochastic Mixup & CutMix* is a practical variant of MSDA by considering Mixup and CutMix at the same time. By a simple alternation of two augmentations, the state-of-the-art performances on large-scale datasets are shown [64, 70]. Stochastic Mixup & CutMix is the same as Equation (1) by setting $M(\lambda) = (1 - \lambda)\vec{1}$ with probability $q$ and $M(\lambda) = M^{\text{cutmix}}(\lambda)$ with probability $1 - q$. We choose $q = 0.5$ as [64, 70]. In our loss function perspective, the regularizing coefficient terms (*i.e.*, $\mathcal{R}_2, \mathcal{R}_3$) become the average of Mixup and CutMix's regularization coefficient. Namely, let $a_{ij}^{\text{mixup}} = (1 - \lambda)^2$ be a regularization coefficient of Mixup and $a_{ij}^{\text{cutmix}}$ be a regularization coefficient of CutMix (10), then the regularization coefficients of Stochastic Mixup & CutMix is $q a_{ij}^{\text{cutmix}} + (1 - q) a_{ij}^{\text{mixup}}$.

Here, we additionally propose two MSDA variants, HMix and GMix, that leverage the advantages of Mixup and CutMix, resulting in showing the intermediate property between Mixup and CutMix.

*Hybrid version of Mixup and CutMix (HMix)* combines Mixup and CutMix by shrinking the CutMix cropped box region and linearly interpolating two images in the areas out of the box as Mixup. The shrinking ratio of the cropped box region is determined by the ratio $r$. HMix can be written as (1) by setting $M$ by (1) randomly cropped box region with side length $\sqrt{1 - \lambda}\sqrt{r}N$ where $N$ is the side length of the original image, and make $M$'s component in the box region as 0 (2) in the areas other than the box, we set $M_i$ as $\frac{\lambda}{1 - (1 - \lambda)r}$. We can easily check that $\mathbb{E}[M] = \lambda\vec{1}$. As $r \to 0$, this method goes to Mixup, and as $r \to 1$, this method goes to CutMix. Note that the ratio $r$ can be a random variable, such as $Beta(\gamma, \gamma)$. In this case, if we set $\gamma \to 0$, as $Beta(\gamma, \gamma)$ goes to Bernoulli distribution, this is equivalent to Stochastic Mixup & CutMix.

We propose *Gaussian Mixup (GMix)* to relax the CutMix box condition to a continuous version as the rectangle cropping of CutMix causes implausible augmented data, *e.g.*, the boundary between two mixed samples. Therefore, we combine two ideas of Mixup and CutMix. Firstly, we select a

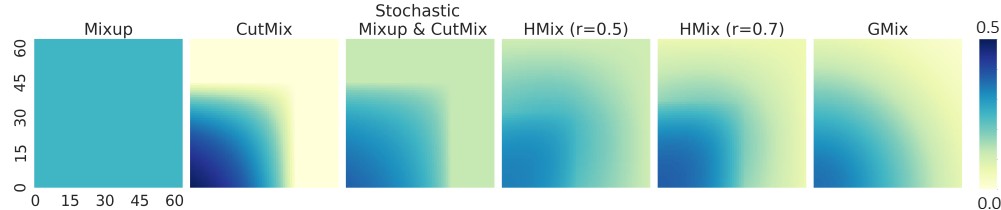

Figure 4: **Visualization of regularization coefficients for different MSDA methods.** $a_{ij}$ values of Mixup, CutMix, Stochastic Mixup & CutMix (the alternation of Mixup and CutMix), HMix, GMix (described in Section 4 and Appendix H) are shown. Each $(x, y)$ value is computed by $\mathbb{E}_i a_{i,i+(x,y)}$ where $i$ is a pixel vector.

point $p$ from the given input. Then, we make $M_i$ as the related function with $\|i - p\|^2$. Specifically, we use the Gaussian function for making $M$: (1) randomly select a point $p$ in image (2) in the areas other than the box, we set $M_i$ as $1 - \exp\left(-\frac{\|i-p\|^2 \pi}{2(1-\lambda)N^2}\right)$. The proposed GMix has the following $a_{ij}$

$$
\begin{aligned}
a_{ij} &= \frac{1}{N^2} \sum_{p \in \text{pixel}} \exp\left(\frac{-\pi}{2(1-\lambda)N^2}\left(-\|i-p\|^2 - \|j-p\|^2\right)\right) \\
&= \int_{\mathbb{R}^2} \exp\left(\frac{-\pi}{2(1-\lambda)N^2}\left(-\|i-p\|^2 - \|j-p\|^2\right)\right) dx \\
&= (1-\lambda) \exp\left(\frac{-\pi}{(1-\lambda)N^2}\left\|\frac{i-j}{2}\right\|^2\right).
\end{aligned}
\tag{11}
$$

As seen in Equation (11), $a_{ij}$ smoothly goes down when the pixel distance becomes larger.

Figure 3 shows the examples generated Mixup, CutMix, HMix, and GMix. The proposed methods (HMix and GMix) generate images in a hybrid form with the properties of both Mixup and CutMix.

**Comparison in terms of regularization coefficients $a_{jk}$.** We illustrate the regularization coefficients $a_{jk}$ of the different MSDA methods in Figure 4. In particular, we fix the mask parameter $\lambda$ to 0.5 and the input resolution to $64 \times 64$. Figure 4 shows the difference between the MSDA methods in terms of how they regularize the input gradients and input Hessians: Mixup has equal weights to every gradient component or Hessian component, while CutMix gives high regularization in close coordinate gradient products or Hessian. We also observe that the hybrid methods (*e.g.*, Stochastic Mixup & CutMix, HMix, and GMix) show the intermediate coefficient values of Mixup and CutMix.

**Comparison in terms of the regularized input gradients after MSDA training.** Equation (8) shows that the regularization term $a_{ij}$ directly affects to the pixel gradients $|\partial_i f_\theta(x_k)\partial_j f_\theta(x_k)|$ in our approximated loss function. The purpose of Figure 5 is to show how the pixel gradients are actually regularized after training. We investigate the amount of the regularized input gradients by $|\partial_v f_\theta(x)\partial_{v+p} f_\theta(x)|$ with respect to the pixel distance vector $p$ for trained models by different MSDA methods. Here, if our approximated loss function actually behaves as a regularization, then we can expect that the pixel gradients $|\partial_v f_\theta(x)\partial_{v+p} f_\theta(x)|$ is small when $a_{ij}$ is large for the given $p$.

We first define the partial gradient product as follows:

$$
\text{PartialGradProd}(x, p) = \max_v |\partial_v f_\theta(x)\partial_{v+p} f_\theta(x)|
\tag{12}
$$

Now, we visualize the pixel-wise maximum values of $\text{PartialGradProd}(x, p)$ in Figure 5. We train different models $f_\theta$ on resized ImageNet (64 x 64) and measure the values on the validation dataset. The $x$-axis and $y$-axis of Figure 5 denote the pixel distance $p$ along each $x$ and $y$ axis, and the scale of the colorbar denotes the value of the maximum partial gradient product. In the figure, we can observe that CutMix reasonably regularizes effectively in the input gradients products when a pixel distance is small; these results aligned with our previous interpretation, CutMix behaves a pixel-level regularizer where it gives stronger regularization (larger $a_{ij}$) to the closer pixels. Note that we are not discussing the relationship between regularizing effects and accuracy but discussing regularizing coefficients and the optimized function's pixel gradients.

**Understanding application cases when a specific MSDA design choice works better than others.** From our theoretical results and empirical studies, we have shown that the design choice of MSDAs

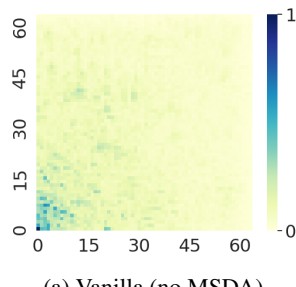 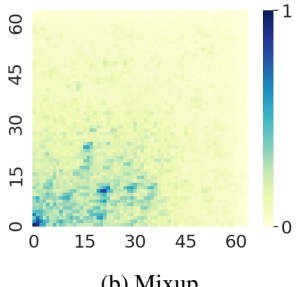 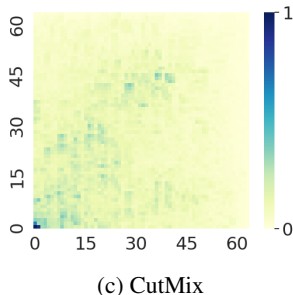

| (a) Vanilla (no MSDA) | (b) Mixup | (c) CutMix |

Figure 5: **Regularized input gradients by MSDA.** The normalized pixel-wise partial gradient norm product comparison among the models trained with vanilla setting (a), Mixup (b) and CutMix (c). We plotted (12), and $x$ and $y$ axis denote the pixel distance $p$ along each axis.

Table 1: **Different tasks need different MSDA strategies.** Validation accuracies of Mixup and CutMix trained networks on two different scenarios on ImageNet-100. Each scenario assumes different pixel importances.

|  | Mixup | CutMix | $\Delta$ (CutMix - Mixup) |
|---|---|---|---|
| Scenario 1: Large crop | 58.3 | **64.4** | **+6.1** |
| Scenario 2: Small crop | **67.7** | 67.0 | **-0.7** |

(*i.e.*, $M$) leads to different regularization effects by regularization coefficient $a_{jk}$. Furthermore, as we have shown in Theorem 2, there always exists a mask that can form any desired $a_{jk}$. We hypothesize that for the given dataset, if a short distance relation is relatively more important than longer distance relations, then CutMix will be better than Mixup. On the contrary, in the opposite case, if a short distance relation is relatively less important, then Mixup will be better than CutMix.

Here, we study different task scenarios when different $a_{jk}$s are required by controlling the pixel-level importance of ImageNet-100 [61] training images. In particular, we design two different scenarios where each of them needs different regularization strategies due to the different pixel-level importance of each task. The results are shown in Table 1. We also report the performance of our proposed methods in both scenarios 1 and 2 in Appendix G.

**Scenario 1: Smaller objects by large crop size.** We randomly crop a large region (80% to 100%) of an image and resize to $64 \times 64$ to train a model. As the objects in the image become small, a close-distance relationship might be more important than a large-distance relationship. Here, we expect CutMix performs better than Mixup as shown in Table 1.

**Scenario 2: Larger objects by small crop size** We randomly crop a small region (25% to 40%) of an image and resize to $64 \times 64$ to train a model. Contrary to Scenario 1, the objects in the image would become large in the cropping region and the large-distance relationship might be important, therefore, we expect that Mixup performs better than CutMix. This hypothesis is aligned to Table 1.

# 5 Comparison of Different MSDA Design Choices: An Empirical Validation

In this section, we compare various MSDA methods on two popular large-scale image classification benchmarks: CIFAR-100 [41] and ImageNet-1K [17]. We will confirm that our proposed design choices, HMix and GMix, are not only theoretically interpolating Mixup and CutMix in the toy settings, but also taking benefits of each method by showing great performances in real-world applications. The implementation details and the hyper-parameter study can be found in Appendix F.

**Results on CIFAR-100 classification.** We evaluate HMix and GMix against baseline MSDA methods including Mixup [76], CutMix [72], Stochastic Mixup & CutMix, [64] and PuzzleMix [39] on CIFAR-100 dataset. Here, we include PuzzleMix, to see the effectiveness of our data-agnostic method against the data-aware mask strategy. Note that although our theoretical results (Section 3) are based on the data-agnostic mask selection methods, our theoretical results can be easily extended to the data-dependent mask selection methods. We leave the extension as a future research direction.

Table 2: **CIFAR-100 classification.** Comparison of various MSDA methods on various network architectures. Note that PuzzleMix needs additional computations (twice than others) for computing the input saliency.

| Augmentation Method | RN56 | WRN28-2 | PreActRN18 | PreActRN34 | PreActRN50 |
|---|---|---|---|---|---|
| Vanilla (no MDSA) | 73.23 | 73.50 | 76.73 | 77.68 | 79.07 |
| Mixup | 73.12 | 74.05 | 77.21 | 79.02 | 79.34 |
| CutMix | 74.83 | 74.79 | 78.66 | 80.05 | 81.23 |
| PuzzleMix | - | 76.51 | 79.38 | 80.89 | 82.46 |
| Stochastic Mixup & CutMix | 74.88 | 75.49 | **79.25** | 81.05 | 81.21 |
| **HMix** (ours) | 74.99 | 75.68 | **79.25** | **81.07** | 81.38 |
| **GMix** (ours) | **75.75** | **76.15** | 79.17 | 80.52 | **81.45** |

To see the generalizability of our methods, we train various network architectures including ResNet-56 (RN56) [26], WideResNet28-2 (WRN28-2) [75], PreActResNet-18 (PreActRN18) [27], PreActResNet-34 (PreActRN34) [27] and PreActResNet-50 (PreActRN50) [27] with various MSDA methods. We train networks for 300 epochs using SGD optimizer with a learning rate 0.2. Table 2 shows the summarized results. We set the hyper-parameter $\alpha$ for Mixup, CutMix, and Stochastic Mixup & CutMix to 1. $\alpha$ for HMix and GMix were set to 1 and 0.5, respectively. We use $r = 0.5$ for HMix. In the table, HMix and GMix outperform Mixup only and CutMix only counterparts and Stochastic Mixup & CutMix often show comparable performances to HMix and GMix. Our methods show comparable performance with the state-of-the-art data-dependent strategy PuzzleMix.

**Results on ImageNet-1K classification.** Table 3 shows the comparison of various MSDA methods on ImageNet-1K. We train ResNet-50 [26] with various MSDA methods for 300 epochs using SGD optimizer with a learning rate 0.1. We set the hyper-parameter $\alpha$ for all methods except Mixup to 1, while Mixup has $\alpha = 0.8$. We use $r = 0.75$ for HMix. Here, we do not include PuzzleMix because it needs heavy additional computations to compute the input saliencies. In the table, HMix shows the best performance, while GMix and Stochastic Mixup & CutMix show the second-best performances. We also include evaluations on various robustness benchmarks in Appendix G.

Table 3: **ImageNet-1K classification.** Comparison of various MSDA methods on ResNet-50 architecture.

| Augmentation Method | Top-1 accuracy |
|---|---|
| Vanilla (no MDSA) | 75.68 (+0.00) |
| Mixup | 77.78 (+2.10) |
| CutMix | 78.04 (+2.36) |
| Stochastic Mixup & CutMix | 78.13 (+2.45) |
| **HMix** (ours) | **78.38** (+2.70) |
| **GMix** (ours) | 78.13 (+2.45) |

## 6 Conclusion

We analyze MSDA by a unified theoretical framework. Our unified theoretical results show that any MSDA method behaves as a regularization on the input gradients and Hessians, where the degree of the regularization is controlled by the design choice of MSDA. We compare various MSDA methods in (1) regularization coefficient (2) regularized gradients (3) model performances in various scenarios with different pixel-level importance. We propose two simple MSDA methods, HMix and GMix, which leverage the benefits of Mixup and CutMix by their design. Our experimental results show that HMix and GMix outperform popular MSDA methods Mixup and CutMix. Furthermore, our methods show comparable or outperformed performances than the state-of-the-art MSDA method, Stochastic Mixup & CutMix, in CIFAR-100 and ImageNet classification tasks.

## Author Contributions

This work is done as an internship project by C Park under the supervision of S Yun. C Park contributed to the theoretical analysis, including theory ideas and proofs. Theoretical results were verified and interpreted by C Park and S Chun. HMix and GMix were designed by C Park and S Yun. S Yun implemented and conducted CIFAR and ImageNet experiments. Empirical findings and analyses were contributed by all authors. Paper presentation, such as paper writing and storyline, is mainly led by S Chun. All authors significantly contributed to the final manuscript.

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
