# Appendix

We include additional materials in this document, including additional theoretical results (Appendix A, B, C, D, E), experimental details (Appendix F), additional experiments (Appendix G), and additional explanation for various MSDAs based on our analysis (Appendix H, I).

## Negative Societal Impacts

Our work focuses on the theoretical aspect of MSDA. Providing a better understanding of data augmentation may help general situations with insufficient data. However, our work does not target a specific scenario that may bring a societal impact. There are no negative societal impacts of our work.

## A    Proof of Theorem 1

For convenience, we restate Theorem 1.

**Theorem 1.** Consider a loss function $l \in \mathcal{L}$. We define $\tilde{D}_\lambda$ as $\frac{\alpha}{\alpha+\beta}\text{Beta}(\alpha+1, \beta) + \frac{\beta}{\alpha+\beta}\text{Beta}(\beta+1, \alpha)$. Assume that $\mathbb{E}_{r_x \sim \mathcal{D}_X}[r_x] = 0$. Then, we can re-write the general MSDA loss (4) as

$$L_m^{\text{MSDA}}(\theta) = L_m(\theta) + \sum_{i=1}^{3} \mathcal{R}_i^{(\text{MSDA})}(\theta) + \mathbb{E}_{\lambda \sim \tilde{\mathcal{D}}(\lambda)}\mathbb{E}_M[(1-M)^\mathsf{T}\varphi(1-M)(1-M)],$$

where $\lim_{a \to 0} \varphi(a) = 0$,

$$\mathcal{R}_1^{(\text{MSDA})}(\theta) = \frac{1}{m}\sum_{i=1}^{m}(h'(f_\theta(x_i)) - y_i)\mathbb{E}_{r_x \sim \mathcal{D}_X}\left(\nabla f_\theta(x_i) \odot (r_x - x_i)\right)^\mathsf{T} \mathbb{E}_{\lambda \sim \tilde{D}_\lambda}\mathbb{E}_M(1-M)$$

$$= \frac{1}{m}\sum_{i=1}^{m}(y_i - h'(f_\theta(x_i)))\left(\nabla f_\theta(x_i)^\mathsf{T} x_i\right)\mathbb{E}_{\lambda \sim \tilde{D}_\lambda}(1-\lambda),$$

$$\mathcal{R}_2^{(\text{MSDA})}(\theta) = \frac{1}{2m}\sum_{i=1}^{m}h''(f_\theta(x_i))\mathbb{E}_{\lambda \sim \tilde{D}_\lambda}\mathbf{G}(\mathcal{D}_X, x_i, f, M),$$

$$\mathcal{R}_3^{(\text{MSDA})}(\theta) = \frac{1}{2m}\sum_{i=1}^{m}(h'(f_\theta(x_i)) - y_i)\mathbb{E}_{\lambda \sim \tilde{D}_\lambda}\mathbf{H}(\mathcal{D}_X, x_i, f, M),$$

and

$$\mathbf{G}(\mathcal{D}_X, x_i, f, M) = \mathbb{E}_M(1-M)^\mathsf{T}\mathbb{E}_{r_x \sim \mathcal{D}_X}\left(\nabla f(x_i) \odot (r_x - x_i)\left(\nabla f(x_i) \odot (r_x - x_i)\right)^\mathsf{T}\right)(1-M)$$

$$= \sum_{j,k \in \text{coord}} a_{jk}\partial_j f_\theta(x_i)\partial_k f_\theta(x_i)\left(\mathbb{E}_{r_x \sim \mathcal{D}_X}[r_{xj}r_{xk}] + x_{ij}x_{ik}\right),$$

$$\mathbf{H}(\mathcal{D}_X, x_i, f, M) = \mathbb{E}_{r_x \sim \mathcal{D}_X}\mathbb{E}_M(1-M)^\mathsf{T}\left(\nabla^2 f_\theta(x_i) \odot ((r_x - x_i)(r_x - x_i)^\mathsf{T})\right)(1-M)$$

$$= \sum_{j,k \in \text{coord}} a_{jk}\left(\mathbb{E}_{r_x \sim \mathcal{D}_X}[r_{xj}r_{xk}\partial_{jk}^2 f_\theta(x_i)] + x_{ij}x_{ik}\partial_{jk}^2 f_\theta(x_i)\right),$$

where

$$a_{jk} := \mathbb{E}_M[(1-M_j)(1-M_k)].$$

*Proof of Theorem 1.* Due to the assumption of the theorem, we can rewrite the empirical loss for the non-augmented population as

$$L_m(\theta) = \frac{1}{m}\sum_{i=1}^{m}l(\theta, z_i) = \frac{1}{m}\sum_{i=1}^{n}[h(f_\theta(x_i)) - y_i f_\theta(x_i)].$$

Similarly, we can rewrite the MSDA loss as

$$L_m^{\text{MSDA}}(\theta) = \frac{1}{m^2}\sum_{i,j=1}^{m}\mathbb{E}_{\lambda \sim \mathcal{D}_\lambda}\mathbb{E}_M l(\theta, \tilde{z}_{i,j}^{(\text{MSDA})}(\lambda, 1-\lambda))$$

$$= \frac{1}{m^2}\sum_{i,j=1}^{m}\mathbb{E}_{\lambda \sim \text{Beta}(\alpha,\beta)}\mathbb{E}_M[h(f_\theta(\tilde{x}_{i,j}^{(\text{MSDA})}(M, 1-M))) - \tilde{y}_{i,j}^{(\text{MSDA})}(\lambda, 1-\lambda)f_\theta(\tilde{x}_{i,j}^{(\text{MSDA})}(M, 1-M))].$$

Putting the definition of $\tilde{z}_{i,j}^{\text{(MSDA)}}(\lambda, 1-\lambda)$ to the equation above, we have

$$L_m^{\text{MSDA}}(\theta) = \frac{1}{m^2} \sum_{i,j=1}^{m} \left( \mathbb{E}_{\lambda \sim \text{Beta}(\alpha,\beta)} \mathbb{E}_M \left( \lambda(h(f_\theta(\tilde{x}_{i,j}^{\text{(MSDA)}}(M, 1-M))) - y_i f_\theta(\tilde{x}_{i,j}^{\text{(MSDA)}}(\lambda, 1-\lambda))) \right. \right.$$

$$\left. \left. + (1-\lambda)(h(f_\theta(\tilde{x}_{i,j}^{\text{(MSDA)}}(M, 1-M))) - y_j f_\theta(\tilde{x}_{i,j}^{\text{(MSDA)}}(M, 1-M))) \right) \right)$$

$$= \frac{1}{m^2} \sum_{i,j=1}^{m} \left( \mathbb{E}_{\lambda \sim \text{Beta}(\alpha,\beta)} \mathbb{E}_{B \sim \text{Bin}(\lambda)} \mathbb{E}_M \left( B(h(f_\theta(\tilde{x}_{i,j}^{\text{(MSDA)}}(M, 1-M))) - y_i f_\theta(\tilde{x}_{i,j}^{\text{(MSDA)}}(M, 1-M))) \right. \right.$$

$$\left. \left. + (1-B)(h(f_\theta(\tilde{x}_{i,j}^{\text{(MSDA)}}(M, 1-M))) - y_j f_\theta(\tilde{x}_{i,j}^{\text{(MSDA)}}(M, 1-M))) \right) \right).$$

Note that $\lambda \sim \text{Beta}(\alpha, \beta)$ and $B|\lambda \sim \text{Bin}(\lambda)$. By conjugacy, we can write the joint distribution of $(\lambda, B)$ as

$$B \sim \text{Bin}\left(\frac{\alpha}{\alpha+\beta}\right), \qquad \lambda|B \sim \text{Beta}(\alpha+B, \beta+1-B).$$

Therefore, we have

$$L_m^{\text{MSDA}}(\theta) = \frac{1}{m^2} \sum_{i,j=1}^{m} \left( \mathbb{E}_{\lambda \sim \text{Beta}(\alpha+1,\beta)} \mathbb{E}_M \frac{\alpha}{\alpha+\beta}(h(f_\theta(\tilde{x}_{i,j}^{\text{(MSDA)}}(\lambda, 1-\lambda))) - y_i f_\theta(\tilde{x}_{i,j}^{\text{(MSDA)}}(\lambda, 1-\lambda))) \right.$$

$$\left. + \mathbb{E}_{\lambda \sim \text{Beta}(\alpha,\beta+1)} \mathbb{E}_M \frac{\beta}{\alpha+\beta}(h(f_\theta(\tilde{x}_{i,j}^{\text{(MSDA)}}(\lambda, 1-\lambda))) - y_j f_\theta(\tilde{x}_{i,j}^{\text{(MSDA)}}(\lambda, 1-\lambda))) \right)$$

$$= \frac{1}{m} \sum_{i=1}^{n} \mathbb{E}_{\lambda \sim \tilde{\mathcal{D}}(\lambda)} \mathbb{E}_{r_x \sim \mathcal{D}_x} \mathbb{E}_M \left[ h\left(f_\theta(M \odot x_i + (1-M) \odot r_x)\right) - y_i f_\theta(M \odot x_i + (1-M) \odot r_x) \right]$$

$$\tag{13}$$

$$= \frac{1}{m} \sum_{i=1}^{n} \mathbb{E}_{\lambda \sim \tilde{\mathcal{D}}(\lambda)} \mathbb{E}_{r_x \sim \mathcal{D}_x} \mathbb{E}_M l(\theta, \hat{z}_i), \tag{14}$$

where $\hat{z}_i = (M \odot x_i + (1-M) \odot r_x, y_i)$.

Let $N = 1 - M$. By defining $\phi_i(N) = h\left(f_\theta(x_i + N \odot (r_x - x_i))\right) - y_i f_\theta(x_i + N \odot (r_x - x_i))$ and applying Taylor expansion, we have

$$\phi_i(N) = \phi_i(0) + \nabla_N \phi_i(0)^\mathsf{T} N + \frac{1}{2} N^\mathsf{T} \nabla_N^2 \phi_i(0) N + N^\mathsf{T} \varphi(N) N, \tag{15}$$

where $\lim_{N \to 0} \varphi(N) = 0$. Firstly, we calculate $\phi_i(0)$ by

$$\phi_i(0) = h(f_\theta(x_i)) - y_i f_\theta(x_i). \tag{16}$$

Second, we calculate $\nabla_N \phi_i(0)$ by

$$\frac{\partial \phi_i(N)}{\partial N_k} = (h'\left(f_\theta\left(x_i + N \odot (r_x - x_i)\right)\right) - y_i) \frac{\partial f_\theta}{\partial x_{ik}}(x_i + N \odot (r_x - x_i))(r_{xk} - x_{ik}),$$

where we denote $N_k$ as the $k$th element of $N$, $x_{ik}$ as the $k$th element of $x_i$, and $r_{xk}$ as the $k$th element of $r_x$. Therefore, we have

$$\nabla_N \phi_i(0)^\mathsf{T} N = (h'(f_\theta(x_i)) - y_i) \sum_k \left( \frac{\partial f_\theta}{\partial x_{ik}}(x_i)(r_{xk} - x_{ik}) \right) N_k$$

$$= (h'(f_\theta(x_i)) - y_i)\left(\nabla f \odot (r_x - x_i)\right) \cdot N. \tag{17}$$

Finally, we calculate $\nabla_N^2 \varphi_i(\vec{0})^T$ by

$$
\begin{aligned}
\frac{\partial^2 \phi_k(N)}{\partial N_k \partial N_j} &= \frac{\partial}{\partial N_j} \left( \left( h'\left( f_\theta\left( x_i + N \odot (r_x - x_i) \right) \right) - y_i \right) \frac{\partial f_\theta}{\partial x_{ik}} \left( x_i + N \odot (r_x - x_i) \right) (r_{xk} - x_{ik}) \right) \\
&= h''\left( f_\theta\left( x_i + N \odot (r_x - x_i) \right) \right) \\
&\quad \times \frac{\partial f_\theta}{\partial x_{ik}} \left( x_i + N \odot (r_x - x_i) \right) (r_{xk} - x_{ik}) \frac{\partial f_\theta}{\partial x_{ij}} \left( x_i + N \odot (r_x - x_i) \right) (r_{xj} - x_{ij}) \\
&\quad + \left( h'\left( f_\theta\left( x_i + N \odot (r_x - x_i) \right) \right) - y_i \right) \\
&\quad \times \frac{\partial^2 f_\theta}{\partial x_{ik} \partial x_{ij}} \left( x_i + N \odot (r_x - x_i) \right) (r_{xk} - x_{ik})(r_{xj} - x_{ij}).
\end{aligned}
$$

Therefore, we have

$$
\begin{aligned}
\frac{1}{2} N^\intercal \nabla_N^2 \phi_i(0) N &= \frac{1}{2} h''\left( f_\theta(x_i) \right) \sum_{k,j} \left( \frac{\partial f_\theta}{\partial x_{ik}}(x_i)(r_{xk} - x_{ik}) \frac{\partial f_\theta}{\partial x_{ij}}(x_i)(r_{xj} - x_{ij}) N_k N_j \right) \\
&\quad + \frac{1}{2} \left( h'\left( f_\theta(x_i) \right) - y_i \right) \sum_{k,j} \frac{\partial^2 f_\theta}{\partial x_{ik} \partial x_{ij}} (x_i) (r_{xk} - x_{ik})(r_{xj} - x_{ij}) N_k N_j \\
&= \frac{1}{2} h''\left( f_\theta(x_i) \right) N^\intercal \left( (\nabla f \odot (r_x - x_i)) (\nabla f \odot (r_x - x_i))^\intercal \right) N \\
&\quad + \frac{1}{2} \left( h'\left( f_\theta(x_i) \right) - y_i \right) N^\intercal \left( \nabla^2 f_\theta(x_i) \odot ((r_x - x_i)(r_x - x_i)^\intercal) \right) N. \quad (18)
\end{aligned}
$$

Applying (16) - (18) to (15),

$$
\begin{aligned}
\phi_i(N) &= \left( h(f_\theta(x_i)) - y_i f_\theta(x_i) \right) + \left( h'(f_\theta(x_i)) - y_i \right) (\nabla f \odot (r_x - x_i)) \cdot N \\
&\quad + \frac{1}{2} h''\left( f_\theta(x_i) \right) N^\intercal \left( (\nabla f \odot (r_x - x_i)) (\nabla f \odot (r_x - x_i))^\intercal \right) N \\
&\quad + \frac{1}{2} \left( h'\left( f_\theta(x_i) \right) - y_i \right) N^\intercal \left( \nabla^2 f_\theta(x_i) \odot ((r_x - x_i)(r_x - x_i)^\intercal) \right) N + N^\intercal \varphi(N) N \quad (19)
\end{aligned}
$$

Plugging (19) to (13), we conclude

$$
\begin{aligned}
L_m^{\text{MSDA}}(\theta) &= \frac{1}{m} \sum_{i=1}^n \mathbb{E}_{\lambda \sim \tilde{\mathcal{D}}(\lambda)} \mathbb{E}_{r_x \sim \mathcal{D}_x} \mathbb{E}_M \phi(1 - M) \\
&= L_m(\theta) + \mathcal{R}_1(\theta) + \mathcal{R}_2(\theta) + \mathcal{R}_3(\theta) + \mathbb{E}_{\lambda \sim \tilde{\mathcal{D}}(\lambda)} \mathbb{E}_M [(1 - M)^\intercal \varphi(1 - M) M],
\end{aligned}
$$

where

$$
\mathcal{R}_1(\theta) = \frac{1}{m} \sum_{i=1}^m (h'(f_\theta(x_i)) - y_i) \left( \nabla f_\theta(x_i) \odot \mathbb{E}_{r_x \sim \mathcal{D}_X} [r_x - x_i] \right) \mathbb{E}_{\lambda \sim \tilde{D}_\lambda} \mathbb{E}_M (1 - M),
$$

$$
\mathcal{R}_2(\theta) = \frac{1}{2m} \sum_{i=1}^m h''(f_\theta(x_i)) \mathbb{E}_{\lambda \sim \tilde{D}_\lambda} \mathbb{E}_M (1 - M)^\intercal \mathbb{E}_{r_x \sim \mathcal{D}_X} \left[ \nabla f(x_i) \odot (r_x - x_i) (\nabla f(x_i) \odot (r_x - x_i))^\intercal \right] (1 - M),
$$

$$
\mathcal{R}_3(\theta) = \frac{1}{2m} \sum_{i=1}^m \mathbb{E}_{\lambda \sim \tilde{D}_\lambda} \mathbb{E}_M (1 - M)^\intercal \mathbb{E}_{r_x \sim \mathcal{D}_X} \left[ \nabla^2 f_\theta(x_i) \odot ((r_x - x_i)(r_x - x_i)^\intercal) \right] (1 - M).
$$

$\square$

# B   Extension of Mixup: $n$-Mixup

In this section, due to notational complexity, we give the approximate loss function of the $n$-sample Mixup ($n$-Mixup). The same analysis can be applied to $n$-sample mixing strategy. We will define $n$-Mixup as followings. Mixup from the $\mathbf{i} = (i_1, i_2, \ldots, i_n)$th samples with $\boldsymbol{\lambda} = (\lambda_1, \lambda_2, \ldots, \lambda_n)$ which is drawn from $\mathcal{D}_\Lambda$ (mainly Dirichlet distribution), is defined as $\tilde{z}_{\mathbf{i}} = \sum_{k=1}^{n} \lambda_k z_{i_k}$. Similarly, we can define the $n$-Mixup loss as

$$L_m^{\text{n-mixup}}(\theta) = \mathbb{E}_{\mathbf{i} \sim \text{Unif}([m])} \mathbb{E}_{\boldsymbol{\lambda} \sim \mathcal{D}_\Lambda} l(\theta, \tilde{z}_{\mathbf{i}}(\boldsymbol{\lambda})) = \frac{1}{m^n} \sum_{\mathbf{i}} \mathbb{E}_{\boldsymbol{\lambda} \sim \mathcal{D}_\Lambda} l(\theta, \tilde{z}_{\mathbf{i}}(\boldsymbol{\lambda})).$$

Throughout this section, we consider $\mathcal{D}_\Lambda$ as Dirichlet distribution (i.e. $\mathcal{D}_\Lambda = \text{Dir}(\boldsymbol{\alpha}) = \text{Dir}(\alpha_1, \alpha_2, \ldots, \alpha_n)$), which is the natural extension of Beta distribution.

**Theorem 5.** *Consider the loss function in $l \in \mathcal{L}$. Then, we can rewrite the n-Mixup loss as*

$$L_m^{\text{n-mix}}(\theta) = \frac{1}{m} \sum_{k=1}^{n} \mathbb{E}_{\boldsymbol{\lambda} \sim \tilde{\mathcal{D}}(\Lambda)} \mathbb{E}_{r_{x,2}, \cdots, r_{x,n} \sim \mathcal{D}_x} \varphi_k(\lambda_2, \cdots, \lambda_n)$$

$$= L_m(\theta) + \mathcal{R}_1(\theta) + \mathcal{R}_2(\theta) + \mathcal{R}_3(\theta) + \mathbb{E}_{\boldsymbol{\lambda} \sim \tilde{\mathcal{D}}(\Lambda)}[o(\|(\lambda_2, \cdots, \lambda_n)\|^2)],$$

*where*

$$\mathcal{R}_1(\theta) = \frac{\mathbb{E}_{\boldsymbol{\lambda} \sim \tilde{\mathcal{D}}_\Lambda}[1 - \lambda_1]}{m} \sum_{i=1}^{m} (h'(f_\theta(x_i)) - y_i) \nabla f_\theta(x_i)^T \mathbb{E}_{r_x \sim \mathcal{D}_X}[r_x - x_i],$$

$$\mathcal{R}_2(\theta) = \frac{\mathbb{E}_{\boldsymbol{\lambda} \sim \tilde{\mathcal{D}}_\Lambda}[\sum_{j=2}^{m} \lambda_j^2]}{2m} \sum_{i=1}^{m} h''(f_\theta(x_i)) \nabla f_\theta(x_i)^T \mathbb{E}_{r_x \sim \mathcal{D}_X}[(r_x - x_i)(r_x - x_i)^T] \nabla_\theta f(x_i)$$

$$+ \frac{\mathbb{E}_{\boldsymbol{\lambda} \sim \tilde{\mathcal{D}}_\Lambda}[(1 - \lambda_1)^2 - \sum_{j=2}^{m} \lambda_j^2]}{2m} \sum_{i=1}^{m} h''(f_\theta(x_i)) \nabla f_\theta(x_i)^T \mathbb{E}_{r_x \sim \mathcal{D}_X}[(r_x - x_i)] \mathbb{E}_{r_x \sim \mathcal{D}_X}[(r_x - x_i)^T] \nabla_\theta f(x_i),$$

$$\mathcal{R}_3(\theta) = \frac{\mathbb{E}_{\boldsymbol{\lambda} \sim \tilde{\mathcal{D}}_\Lambda}[\sum_{j=2}^{m} \lambda_j^2]}{2m} \sum_{i=1}^{m} (h'(f_\theta(x_i)) - y_i) \mathbb{E}_{r_x \sim \mathcal{D}_X}[(r_x - x_i) \nabla^2 f_\theta(x_i)(r_x - x_i)^T]$$

$$+ \frac{\mathbb{E}_{\boldsymbol{\lambda} \sim \tilde{\mathcal{D}}_\Lambda}[(1 - \lambda_1)^2 - \sum_{j=2}^{m} \lambda_j^2]}{2m} \sum_{i=1}^{m} (h'(f_\theta(x_i)) - y_i) \mathbb{E}_{r_x \sim \mathcal{D}_X}[(r_x - x_i)] \nabla^2 f_\theta(x_i) \mathbb{E}_{r_x \sim \mathcal{D}_X}[(r_x - x_i)^T].$$

As Mixup's approximate loss function [77] or Theorem 1, $n$-Mixup also regularizes $\nabla f$ and $\nabla^2 f$.

*Proof.* Due to the assumption of the theorem, we can rewrite the empirical loss for the non-augmented population as

$$L_m(\theta) = \frac{1}{m} \sum_{i=1}^{m} l(\theta, z_i) = \frac{1}{m} \sum_{i=1}^{n} [h(f_\theta(x_i)) - y_i f_\theta(x_i)].$$

Similarly, we can rewrite the $n$-Mixup loss as

$$L_m^{\text{n-mix}}(\theta) = \frac{1}{m^n} \sum_{\mathbf{i}} \mathbb{E}_{\boldsymbol{\lambda} \sim \mathcal{D}_\Lambda} l(\theta, \tilde{z}_{\mathbf{i}}(\boldsymbol{\lambda})) = \frac{1}{m^n} \mathbb{E}_{\boldsymbol{\lambda} \sim \text{Dir}(\boldsymbol{\alpha})} \sum_{\mathbf{i}} [h(f_\theta(\tilde{x}_{\mathbf{i}}(\boldsymbol{\lambda}))) - \tilde{y}_{\mathbf{i}}(\boldsymbol{\lambda}) f_\theta(\tilde{x}_{\mathbf{i}}(\boldsymbol{\lambda}))].$$

Putting the definition of $\tilde{x}_{\mathbf{i}}$ and $\tilde{y}_{\mathbf{i}}$ to the equation above, we have

$$L_m^{\text{n-mix}}(\theta) = \frac{1}{m^n} \mathbb{E}_{\boldsymbol{\lambda} \sim \text{Dir}(\boldsymbol{\alpha})} \sum_{\mathbf{i}} \sum_{k=1}^{n} \lambda_k (h(f_\theta(\tilde{x}_{\mathbf{i}}(\boldsymbol{\lambda}))) - y_k f_\theta(\tilde{x}_{\mathbf{i}}(\boldsymbol{\lambda})))$$

$$= \frac{1}{m^n} \sum_{\mathbf{i}} \mathbb{E}_{\boldsymbol{\lambda} \sim \text{Dir}(\boldsymbol{\alpha})} \mathbb{E}_{\boldsymbol{\beta} \sim \text{Mult}(\boldsymbol{\lambda})} \sum_{k=1}^{n} \beta_k (h(f_\theta(\tilde{x}_{\mathbf{i}}(\boldsymbol{\lambda}))) - y_k f_\theta(\tilde{x}_{\mathbf{i}}(\boldsymbol{\lambda}))),$$

where $\boldsymbol{\beta} = (\beta_1, \beta_2, \cdots, \beta_n)$ and $\text{Mult}(\boldsymbol{\lambda})$ is multinomial distribution. Note that $\boldsymbol{\lambda} \sim \text{Dir}(\boldsymbol{\alpha})$ and $\boldsymbol{\beta}|\boldsymbol{\lambda} \sim \text{Mult}(\boldsymbol{\lambda})$. By conjugacy, we can write the joint distribution of $(\boldsymbol{\alpha}, \boldsymbol{\beta})$ as

$$\boldsymbol{\beta} \sim \text{Mult}\left(\frac{\boldsymbol{\alpha}}{\sum \alpha_i}\right), \qquad \boldsymbol{\lambda}|\boldsymbol{\beta} \sim \text{Dir}(\boldsymbol{\alpha} + \boldsymbol{\beta}).$$

Therefore,

$$L_m^{\text{n-mix}}(\theta) = \frac{1}{m^n} \sum_{\mathbf{i}} \sum_{k=1}^{n} \frac{\alpha_k}{\sum \alpha_i} \mathbb{E}_{\boldsymbol{\lambda} \sim \text{Dir}(\alpha_1, \alpha_2, \ldots, \alpha_k+1, \ldots, \alpha_n)} (h(f_\theta(\tilde{x}_{\mathbf{i}}(\boldsymbol{\lambda}))) - y_k f_\theta(\tilde{x}_{\mathbf{i}}(\boldsymbol{\lambda})))$$

$$= \frac{1}{m} \sum_{k=1}^{n} \mathbb{E}_{\boldsymbol{\lambda} \sim \tilde{\mathcal{D}}(\Lambda)} \mathbb{E}_{r_{x,2}, \cdots, r_{x,n} \sim \mathcal{D}_x} \left[ h\left(f_\theta\left(\lambda_1 x_k + \sum_{j=2}^{n} \lambda_j r_{x,j}\right)\right) - y_k f_\theta\left(\lambda_1 x_k + \sum_{j=2}^{n} \lambda_j r_{x,j}\right)\right],$$
(20)

where $\tilde{D}_\Lambda = \sum_{l=1}^{n} \frac{\alpha_l}{\sum \alpha_i} \text{Dir}(\alpha_l+1, \alpha_{l+1}, \ldots, \alpha_{l+n-1})$ and $\mathcal{D}_x$ is the empirical distribution induced by training samples. We regard the index with mod $n$. Defining $\varphi_k(\cdot)$ as

$$\varphi_k(\lambda_2, \cdots, \lambda_n) = h\left(f_\theta\left((1 - \sum_{j=2}^{n} \lambda_j)x_k + \sum_{j=2}^{n} \lambda_j r_{x,j}\right)\right) - y_k f_\theta\left((1 - \sum_{j=2}^{n} \lambda_j)x_k + \sum_{j=2}^{n} \lambda_j r_{x,j}\right),$$

we can use twice the differentiability of $f(\cdot)$ and $h(\cdot)$, so we have

$$\varphi_k(\lambda_2, \cdots, \lambda_n) = \varphi_k(\vec{0}) + \nabla\varphi_k(\vec{0})^T (\lambda_2, \cdots, \lambda_n) + \frac{1}{2}(\lambda_2, \cdots, \lambda_n)^T \nabla^2\varphi_k(\vec{0})^T (\lambda_2, \cdots, \lambda_n)^T + o(\|(\lambda_2, \cdots, \lambda_n)\|^2).$$
(21)

Firstly, we calculate $\varphi_k(\vec{0})$ by

$$\varphi_k(\vec{0}) = h(f_\theta(x_k)) - y_k f_\theta(x_k).$$
(22)

Second, we calculate $\nabla\varphi_k(\vec{0})$ by

$$\frac{\partial\varphi_k(\lambda_2, \cdots, \lambda_n)}{\partial\lambda_i} = h'\left(f_\theta\left((1 - \sum_{j=2}^{n} \lambda_j)x_k + \sum_{j=2}^{n} \lambda_j r_{x,j}\right)\right) f_\theta'\left((1 - \sum_{j=2}^{n} \lambda_j)x_k + \sum_{j=2}^{n} \lambda_j r_{x,j}\right)(r_{x,i} - x_k)$$

$$- y_k f_\theta'\left((1 - \sum_{j=2}^{n} \lambda_j)x_k + \sum_{j=2}^{n} \lambda_j r_{x,j}\right)(r_{x,i} - x_k)$$

Therefore, we have

$$\left.\frac{\partial\varphi_k(\lambda_2, \cdots, \lambda_n)}{\partial\lambda_i}\right|_{(\lambda_2, \cdots, \lambda_n) = \vec{0}} = (h'(f_\theta(x_k)) - y_i)\nabla f_\theta(x_k)^T (r_{x,i} - x_k).$$
(23)

Finally, we calculate $\nabla^2\varphi_i(\vec{0})^T$ by

$$\frac{\partial^2\varphi_k(\lambda_2, \cdots, \lambda_n)}{\partial\lambda_i \partial\lambda_s} = \frac{\partial}{\partial\lambda_s}\left(h'\left(f_\theta\left((1 - \sum_{j=2}^{n} \lambda_j)x_k + \sum_{j=2}^{n} \lambda_j r_{x,j}\right)\right) f_\theta'\left((1 - \sum_{j=2}^{n} \lambda_j)x_k + \sum_{j=2}^{n} \lambda_j r_{x,j}\right)(r_{x,i} - x_k)\right.$$

$$\left. - y_k f_\theta'\left((1 - \sum_{j=2}^{n} \lambda_j)x_k + \sum_{j=2}^{n} \lambda_j r_{x,j}\right)(r_{x,i} - x_k)\right)$$

$$= h''\left(f_\theta\left((1 - \sum_{j=2}^{n} \lambda_j)x_k + \sum_{j=2}^{n} \lambda_j r_{x,j}\right)\right)\left[f_\theta'\left((1 - \sum_{j=2}^{n} \lambda_j)x_k + \sum_{j=2}^{n} \lambda_j r_{x,j}\right)(r_{x,i} - x_k)\right]$$

$$\left[f_\theta'\left((1 - \sum_{j=2}^{n} \lambda_j)x_k + \sum_{j=2}^{n} \lambda_j r_{x,j}\right)(r_{x,s} - x_k)\right]$$

$$+ h'\left(f_\theta\left((1 - \sum_{j=2}^{n} \lambda_j)x_k + \sum_{j=2}^{n} \lambda_j r_{x,j}\right)\right)(r_{x,i} - x_k)^T \nabla^2 f_\theta\left((1 - \sum_{j=2}^{n} \lambda_j)x_k + \sum_{j=2}^{n} \lambda_j r_{x,j}\right)(r_{x,s} - x_k)$$

$$- y_k (r_{x,i} - x_k)^T \nabla^2 f_\theta\left((1 - \sum_{j=2}^{n} \lambda_j)x_k + \sum_{j=2}^{n} \lambda_j r_{x,j}\right)(r_{x,s} - x_k).$$

Therefore,

$$\frac{\partial^2 \varphi_k(\lambda_2, \cdots, \lambda_n)}{\partial \lambda_i \partial \lambda_s}\bigg|_{(\lambda_2, \cdots, \lambda_n) = \vec{0}} = (h''(f_\theta(x_k) - y_k))\left(\nabla f_\theta(x_k)^T (r_{x,i} - x_k)\right)\left(\nabla f_\theta(x_k)^T (r_{x,s} - x_k)\right)$$

$$- y_k (r_{x,i} - x_k)^T \nabla^2 f_\theta(x_k)(r_{x,s} - x_k). \tag{24}$$

Applying (22) - (24) to (21),

$$\varphi_k(\lambda_2, \cdots, \lambda_n) = (h(f_\theta(x_k)) - y_k f_\theta(x_k)) + \sum_{j=2}^{n}(h'(f_\theta(x_k)) - y_i)\nabla f_\theta(x_k)^T (r_{x,i} - x_k)\lambda_j$$

$$+ \frac{1}{2}\sum_{i,s=2}^{n}\left((h''(f_\theta(x_k) - y_k))\left(\nabla f_\theta(x_k)^T (r_{x,i} - x_k)\right)\left(\nabla f_\theta(x_k)^T (r_{x,s} - x_k)\right)\right.$$

$$\left. - y_k(r_{x,i} - x_k)^T \nabla^2 f_\theta(x_k)(r_{x,s} - x_k)\right)\lambda_i \lambda_s + o(\|(\lambda_2, \cdots, \lambda_n)\|^2). \tag{25}$$

Plugging (25) to (20),

$$L_m^{\text{n-mix}}(\theta) = \frac{1}{m}\sum_{k=1}^{n}\mathbb{E}_{\boldsymbol{\lambda} \sim \tilde{\mathcal{D}}(\Lambda)}\mathbb{E}_{r_{x,2}, \cdots, r_{x,n} \sim \mathcal{D}_x}\varphi_k(\lambda_2, \cdots, \lambda_n)$$

$$= L_m(\theta) + \mathcal{R}_1(\theta), + \mathcal{R}_2(\theta) + \mathcal{R}_3(\theta) + \mathbb{E}_{\boldsymbol{\lambda} \sim \tilde{\mathcal{D}}(\Lambda)}[o(\|(\lambda_2, \cdots, \lambda_n)\|^2)],$$

where

$$\mathcal{R}_1(\theta) = \frac{\mathbb{E}_{\boldsymbol{\lambda} \sim \tilde{\mathcal{D}}_\Lambda}[1 - \lambda_1]}{m}\sum_{i=1}^{m}(h'(f_\theta(x_i)) - y_i)\nabla f_\theta(x_i)^T \mathbb{E}_{r_x \sim \mathcal{D}_X}[r_x - x_i],$$

$$\mathcal{R}_2(\theta) = \frac{\mathbb{E}_{\boldsymbol{\lambda} \sim \tilde{\mathcal{D}}_\Lambda}[\sum_{j=2}^{m}\lambda_j^2]}{2m}\sum_{i=1}^{m}h''(f_\theta(x_i))\nabla f_\theta(x_i)^T \mathbb{E}_{r_x \sim \mathcal{D}_X}[(r_x - x_i)(r_x - x_i)^T]\nabla_\theta f(x_i)$$

$$+ \frac{\mathbb{E}_{\boldsymbol{\lambda} \sim \tilde{\mathcal{D}}_\Lambda}[(1 - \lambda_1)^2 - \sum_{j=2}^{m}\lambda_j^2]}{2m}\sum_{i=1}^{m}h''(f_\theta(x_i))\nabla f_\theta(x_i)^T \mathbb{E}_{r_x \sim \mathcal{D}_X}[(r_x - x_i)]\mathbb{E}_{r_x \sim \mathcal{D}_X}[(r_x - x_i)^T]\nabla_\theta f(x_i),$$

$$\mathcal{R}_3(\theta) = \frac{\mathbb{E}_{\boldsymbol{\lambda} \sim \tilde{\mathcal{D}}_\Lambda}[\sum_{j=2}^{m}\lambda_j^2]}{2m}\sum_{i=1}^{m}(h'(f_\theta(x_i)) - y_i)\mathbb{E}_{r_x \sim \mathcal{D}_X}[(r_x - x_i)\nabla^2 f_\theta(x_i)(r_x - x_i)^T]$$

$$+ \frac{\mathbb{E}_{\boldsymbol{\lambda} \sim \tilde{\mathcal{D}}_\Lambda}[(1 - \lambda_1)^2 - \sum_{j=2}^{m}\lambda_j^2]}{2m}\sum_{i=1}^{m}(h'(f_\theta(x_i)) - y_i)\mathbb{E}_{r_x \sim \mathcal{D}_X}[(r_x - x_i)]\nabla^2 f_\theta(x_i)\mathbb{E}_{r_x \sim \mathcal{D}_X}[(r_x - x_i)^T].$$

$$\square$$

## C  Adversarial Robustness of MSDA

Let us scrutinize adversarial robustness in MSDA. We adopt the logistic loss, so $l(\theta, z) = \log(1 + \exp(f_\theta(x))) - yf_\theta(x)$ where $y \in \{0, 1\}$. Define $g(s) = e^s/(1 + e^s)$. As [77], we scrutinize the logistic regression with $f_\theta(x)$ as ReLU or leaky-ReLU network. Then, we have $f_\theta(x) = \nabla f_\theta(x_i)^\intercal x_i$ and $\nabla^2 f_\theta(x_i) = 0$. We consider the adversarial loss with $l_2$ attack of size $\epsilon\sqrt{d}$ ($d$ is the dimension of $\theta$), that is, $L_m^{\text{adv}}(\theta) = \frac{1}{m}\sum_{i=1}^{m}\max_{\|\delta_i\|_2 \le \epsilon\sqrt{d}} l(\theta, (x_i + \delta_i, y_i))$.

**Theorem 3.** *In MSDA, we suppose that $f_\theta(x) = \nabla f_\theta(x_i)^\intercal x_i, \nabla^2 f_\theta(x_i) = 0$ and there exists a constant $c_x > 0$ that $\|x_i\|_2 \le c_x\sqrt{d}$ for all $i \in [m]$. Then for any $\theta \in \Theta$, we have*

$$\tilde{L}_m^{(MSDA)} \ge \frac{1}{m}\sum_{i=1}^{m}\tilde{l}_{adv}(\epsilon_i\sqrt{d}, z) \ge \frac{1}{m}\sum_{i=1}^{m}\tilde{l}_{adv}(\epsilon_{cut}\sqrt{d}, z),$$

where $\epsilon_{cut} = c_x \min \left( \min_i |\cos(\nabla f_\theta(x_i), x_i)| \mathbb{E}_{\lambda \sim \tilde{\mathcal{D}}_\lambda}[1-\lambda], \min_i \mathbb{E}_{\lambda \sim \tilde{D}_\lambda, M} s(M, x_i) \right)$ and $s(M, x_i) = \frac{(\sqrt{1-M} \odot \nabla f(x_i))^T (\sqrt{1-M} \odot x_i)}{\|\nabla f(x_i)\|_2 \|x_i\|_2}$.

This bound appears to be fertile at first glance. However, as $\|f_\theta(x_i)\|_2$ increases after training accuracy reaches 100%, the logistic loss decreases. Therefore, due to $\|f_\theta(x_i)\|_2 = \|\nabla f_\theta(x_i)^\intercal x_i\|_2 = \|\nabla f_\theta(x_i)\|_2 \|x_i\|_2 \cos(\nabla f_\theta(x_i), x_i)$, $\cos(\nabla f_\theta(x_i), x_i)$ would be larger. Furthermore, under the CutMix case, since $M$ distribution is relatively uniform, $s(M, x_i)$ will be similar to $\min_i |\cos(\nabla f_\theta(x_i), x_i)| \mathbb{E}_{\lambda \sim \tilde{\mathcal{D}}_\lambda}[1-\lambda]$ [77]. Moreover, empirical results support Mixup's or Cutmix's adversarial robustness [76, 72, 55].

*proof of Theorem 3.*

**Fact 1.** [77] *The second order Taylor approximation of $L_m^{adv}(\theta)$ is $\frac{1}{m} \sum_{i=1}^m \tilde{l}_{adv}(\epsilon\sqrt{d}, z)$ where fore any $\eta > 0, x \in \mathbb{R}^d$ and $y \in \{0, 1\}$,*

$$\tilde{l}_{adv}(\eta, z) = l(\theta, z) + \eta|g(f_\theta(x)) - y| \|\nabla f_\theta(x)\|_2 + \frac{\eta^2 d}{2} \cdot |h''(f_\theta(x))| \cdot \|\nabla f_\theta(x)\|_2^2.$$

We set $\mathbb{E}_{r_x}(r_x) = 0$ by parallel translation. We define

$$s(M, x_i) = \frac{(\sqrt{1-M} \odot \nabla f(x_i))^T (\sqrt{1-M} \odot x_i)}{\|\nabla f(x_i)\|_2 \|x_i\|_2}.$$

For every MSDA, we have $\mathcal{R}_1$ as

$$\mathcal{R}_1(\theta) = \frac{\mathbb{E}_\lambda(1-\lambda)}{m} \sum_{i=1}^m (y_i - g(f_\theta(x_i))) f_\theta(x_i),$$

and since $\theta \in \Theta$, we have $(y_i - g(f_\theta(x_i))) f_\theta(x_i) \geq 0$. Therefore,

$$\mathcal{R}_1(\theta) = \frac{\mathbb{E}_\lambda(1-\lambda)}{m} \sum_{i=1}^m |y_i - g(f_\theta(x_i))||f_\theta(x_i)|$$

$$= \frac{\mathbb{E}_\lambda(1-\lambda)}{m} \sum_{i=1}^m |y_i - g(f_\theta(x_i))| \|\nabla f_\theta(x_i)\|_2 \|x_i\|_2 |\cos(\nabla f_\theta(x_i), x_i)|$$

$$\geq \frac{\min_i \|x_i\|_2 \min_i |\cos(\nabla f_\theta(x_i), x_i)| \mathbb{E}_\lambda(1-\lambda)}{m} \sum_{i=1}^m |y_i - g(f_\theta(x_i))| \|\nabla f_\theta(x_i)\|_2.$$

Moreover, we can eliminate $\mathcal{R}_3$ since $\nabla^2 f_\theta = 0$. So, we only focus on $\mathcal{R}_2$ term. We have

$$\mathcal{R}_2^{(\text{MSDA})}(\theta) = \frac{1}{2m} \sum_{i=1}^m h''(f_\theta(x_i)) \mathbb{E}_{\lambda \sim \tilde{D}_\lambda} \mathbb{E}_M (1-M)^\intercal \mathbb{E}_{r_x \sim \mathcal{D}_X} (\nabla f(x_i) \odot (r_x - x_i) (\nabla f(x_i) \odot (r_x - x_i))^\intercal) (1-M)$$

$$\geq \frac{1}{2m} \sum_{i=1}^m h''(f_\theta(x_i)) \mathbb{E}_{\lambda \sim \tilde{D}_\lambda} \mathbb{E}_M (1-M)^\intercal ((\nabla f(x_i) \odot x_i) (\nabla f(x_i) \odot x_i)^\intercal) (1-M)$$

$$= \frac{1}{2m} \sum_{i=1}^m |g(f_\theta(x_i))(1 - g(f_\theta(x_i)))| \mathbb{E}_{\lambda \sim \tilde{D}_\lambda} \mathbb{E}_M ((\sqrt{1-M} \odot \nabla f(x_i))^T (\sqrt{1-M} \odot x_i))^2$$

$$= \frac{1}{2m} \sum_{i=1}^m |g(f_\theta(x_i))(1 - g(f_\theta(x_i)))| \mathbb{E}_{\lambda \sim \tilde{D}_\lambda, M} s(M, x_i)^2 \|\nabla f(x_i)\|_2^2 \|x_i\|_2^2$$

$$\geq \frac{\min_i \|x_i\|^2 \min_i \mathbb{E}_{\lambda \sim \tilde{D}_\lambda, M} s(M, x_i)^2}{2m} \sum_{i=1}^m |g(f_\theta(x_i))(1 - g(f_\theta(x_i)))| \|\nabla f(x_i)\|_2^2,$$

which concludes the theorem. $\square$

# D  Generalization properties of MSDA

The data-dependent MSDA regularization can be altered to the original empirical risk minimization problem with a constrained function set. The Rademacher complexity of this constrained function set is $\mathcal{O}(1/\sqrt{n})$, which leads to the generalization properties of MSDA. We investigate two models. The first is the GLM model, which has the loss function $l(\theta, z) = A(\theta^{\mathsf{T}}x) - y\theta^{\mathsf{T}}x$. The second is two-layer ReLU networks, which can be parameterized as $f_\theta(x) = \theta_1^{\mathsf{T}}\sigma(Wx) + \theta_0$. In this case, we consider the mean square error (MSE) loss function (*i.e.*, $l(\theta) = \frac{1}{m}\sum_{i=1}^m (y_i - f_\theta(x_i))^2$)

## D.1  GLM Model

For GLM, using (14), since the prediction of the GLM model is invariant to the scaling of the training data, we think the dataset $\hat{D} = \{\hat{z}_i\}_{i=1}^m$ with $\hat{x}_i = 1 \oslash \bar{M} \odot (M \odot x_i + (1 - M) \odot r_x)$ where $\bar{M} = \mathbb{E}M$. Then, the loss function is

$$L_m^{(\text{MSDA})} = \frac{1}{m}\mathbb{E}_\lambda \mathbb{E}_{r_x} \mathbb{E}_M \sum_{i=1}^m l(\theta, \tilde{z}_i) = \frac{1}{m}\mathbb{E}_\xi \sum_{i=1}^m (A(\hat{x}_i^{\mathsf{T}}\theta) - y_i\hat{x}_i^{\mathsf{T}}\theta),$$

where $\xi$ denotes the randomness of $\lambda, r_x$, and $M$. By the second approximation of $A(\cdot)$, we can express $A(\hat{x}_i^{\mathsf{T}}\theta)$ as

$$A(\hat{x}_i^{\mathsf{T}}\theta) = A(x_i^{\mathsf{T}}\theta) + A'(x_i^{\mathsf{T}}\theta)(\hat{x}_i - x_i)^{\mathsf{T}}\theta + \frac{1}{2}A''(x_i^{\mathsf{T}}\theta)\theta^{\mathsf{T}}(\hat{x}_i - x_i)(\hat{x}_i - x_i)^{\mathsf{T}}\theta$$

to approximate the loss function. Therefore, we have

$$\tilde{L}_m^{(\text{MSDA})} = \frac{1}{m}\sum_{i=1}^m A(x_i^{\mathsf{T}}\theta) + \frac{1}{m}\mathbb{E}_\xi \sum_{i=1}^m \left( A'(x_i^{\mathsf{T}}\theta)(\hat{x}_i - x_i)^{\mathsf{T}}\theta + \frac{1}{2}A''(x_i^{\mathsf{T}}\theta)\theta^{\mathsf{T}}(\hat{x}_i - x_i)(\hat{x}_i - x_i)^{\mathsf{T}}\theta \right)$$

$$= \frac{1}{m}\sum_{i=1}^m A(x_i^{\mathsf{T}}\theta) + \frac{1}{m}\sum_{i=1}^m \left( \frac{1}{2}A''(x_i^{\mathsf{T}}\theta)\theta^{\mathsf{T}}\text{Var}_\xi(\hat{x}_i)\theta \right), \tag{26}$$

where $\tilde{L}_m^{(\text{MSDA})}$ denotes the approximate loss of $L_m^{(\text{MSDA})}$ since $\mathbb{E}_\xi r_x = 0$ and $\mathbb{E}_\xi \hat{x}_i = x_i$. For calculating $\text{Var}_\xi(\hat{x}_i)$, we use the law of total variance. We have

$$\text{Var}_\xi(\tilde{x}_i) = \left( \frac{1}{\bar{M}} \frac{1}{\bar{M}}^{\mathsf{T}} \right) \odot \text{Var}_\xi (M \odot x_i + (1 - M) \odot r_x)$$

$$= \left( \frac{1}{\bar{M}} \frac{1}{\bar{M}}^{\mathsf{T}} \right) \odot (\mathbb{E}(\text{Var}(M \odot x_i + (1 - M) \odot r_x \,|\, \lambda, M) + \text{Var}(\mathbb{E}(M \odot x_i + (1 - M) \odot r_x \,|\, \lambda, M))$$

$$= \left( \frac{1}{\bar{M}} \frac{1}{\bar{M}}^{\mathsf{T}} \right) \odot \left( \mathbb{E}(1 - M)\hat{\Sigma}_X(1 - M)^{\mathsf{T}} + x_i\text{Var}(M)x_i^{\mathsf{T}} \right)$$

$$= \frac{1}{\bar{\lambda}^2} \left( \mathbb{E}(1 - M)\hat{\Sigma}_X(1 - M)^{\mathsf{T}} + x_i\text{Var}(M)x_i^{\mathsf{T}} \right),$$

where $\hat{\Sigma}_X = \frac{1}{m}\sum_{i=1}^m x_i x_i^{\mathsf{T}}$ with some notational ambiguity that $\frac{1}{\bar{M}} := \vec{1} \oslash \bar{M}$. In our setting $\bar{M} = \bar{\lambda}\vec{1}$ where $\bar{\lambda} = \mathbb{E}_{\lambda \sim \tilde{\mathcal{D}}_\lambda}[\lambda]$. Now we think the related dual problem to the (26):

$$\mathcal{W}_\gamma = \left\{ x \to \theta^{\mathsf{T}}x, \text{ such that } \theta \text{ satisfying} \right.$$

$$\left. (\mathbb{E}_x A''(\theta^{\mathsf{T}}x)) \cdot (\theta^{\mathsf{T}}(\mathbb{E}(1 - M)\Sigma_X(1 - M)^{\mathsf{T}})\theta + \theta^{\mathsf{T}}((x\text{Var}(M)x^{\mathsf{T}}))\theta) \leq \gamma \right\}.$$

Here, we assume the $(\mathbb{E}_x[A''(v^{\mathsf{T}}x)])^2 \geq \rho\mathbb{E}_x(v^{\mathsf{T}}x)^2$, which is called $\rho$-retentiveness [77, 1].

**Theorem 4-(a)** (Restated). *Define $\Sigma_X^{(M)} = \mathbb{E}(1 - M)\Sigma_X(1 - M)^{\mathsf{T}}$. Suppose $A(\cdot)$ is $L_A$ Lipschitz, and $\mathcal{X}, \mathcal{Y}, \Theta$ are all bounded. There exist constants $L, B > 0$, such that for all $\theta$ that $\theta^{\mathsf{T}}x \in \mathcal{W}_\gamma$, which is the regularization induced by MSDA, we have*

$$L(\theta) \leq L_m(\theta) + 2LL_A\frac{1}{\sqrt{n}}(\gamma/\rho)^{1/4} \left( \sqrt{tr\left( \left(\Sigma_X^{(M)}\right)^\dagger \Sigma_X \right)} + rank(\Sigma_X) \right) + B\sqrt{\frac{\log(1/\delta)}{2n}},$$

*with probability at least $1 - \delta$.*

*Proof.* Firstly, we calculate the empirical Rademacher complexity of $\mathcal{W}_\gamma$. For $n$ i.i.d. Rademacher random variables $\xi_1, \ldots, \xi_n$, the definition of the empirical Rademacher complexity gives

$$\text{Rad}(\mathcal{W}_\gamma, n) = \mathbb{E}_{\xi_i} \sup_{(\mathbb{E}_x A''(\theta^\mathsf{T} x)) \cdot \left(\theta^\mathsf{T} \Sigma_X^{(M)} \theta + \theta^\mathsf{T}((x\text{Var}(M)x^\mathsf{T}))\right) \theta \leq \gamma} \frac{1}{n} \sum_{i=1}^n \xi_i \theta^\mathsf{T} x_i$$

$$\leq \mathbb{E}_{\xi_i} \sup_{(\mathbb{E}_x A''(\theta^\mathsf{T} x)) \cdot \theta^\mathsf{T} \Sigma_X^{(M)} \theta \leq \gamma} \frac{1}{n} \sum_{i=1}^n \xi_i \theta^\mathsf{T} x_i.$$

Due to the $\rho$-retentiveness, we have

$$\text{Rad}(\mathcal{W}_\gamma, n) \leq \mathbb{E}_{\xi_i} \sup_{(\theta^\mathsf{T} \Sigma_X \theta) \cdot \left(\theta^\mathsf{T} \Sigma_X^{(M)} \theta\right) \leq \gamma/\rho} \frac{1}{n} \sum_{i=1}^n \xi_i \theta^\mathsf{T} x_i$$

$$\leq \mathbb{E}_{\xi_i} \left( \sup_{\theta^\mathsf{T} \Sigma_X \theta \leq \sqrt{\gamma/\rho}} \frac{1}{n} \sum_{i=1}^n \xi_i \theta^\mathsf{T} x_i + \mathbb{E}_{\xi_i} \sup_{\theta^\mathsf{T} \Sigma_X^{(M)} \theta \leq \sqrt{\gamma/\rho}} \frac{1}{n} \sum_{i=1}^n \xi_i \theta^\mathsf{T} x_i \right)$$

For the first part of the RHS, define $\tilde{x}_i = \Sigma_X^{\dagger/2} x_i$ and $v = \Sigma_X^{1/2} \theta$. Then, we have

$$\mathbb{E}_{\xi_i} \sup_{\theta^\mathsf{T} \Sigma_X \theta \leq \sqrt{\gamma/\rho}} \frac{1}{n} \sum_{i=1}^n \xi_i \theta^\mathsf{T} x_i = \mathbb{E}_{\xi_i} \sup_{\|v\|^2 \leq \sqrt{\gamma/\rho}} \frac{1}{n} \sum_{i=1}^n \xi_i v^\mathsf{T} \tilde{x}_i$$

$$\leq \frac{1}{n} (\gamma/\rho)^{1/4} \mathbb{E}_{\xi_i} \left\| \sum_{i=1}^n \xi_i \tilde{x}_i \right\| \leq \frac{1}{n} (\gamma/\rho)^{1/4} \sqrt{\mathbb{E}_{\xi_i} \left\| \sum_{i=1}^n \xi_i \tilde{x}_i \right\|^2}$$

$$= \frac{1}{n} (\gamma/\rho)^{1/4} \sqrt{\sum_{i=1}^n \tilde{x}_i^\mathsf{T} \tilde{x}_i}.$$

Similarly, by defining $\check{x}_i = \left(\Sigma_X^{(M)}\right)^{\dagger/2} x_i$ and $v = \left(\Sigma_X^{(M)}\right)^{1/2} \theta$,

$$\mathbb{E}_{\xi_i} \sup_{\theta^\mathsf{T} \Sigma_X^{(M)} \theta \leq \sqrt{\gamma/\rho}} \frac{1}{n} \sum_{i=1}^n \xi_i \theta^\mathsf{T} x_i = \mathbb{E}_{\xi_i} \sup_{\|v\|^2 \leq \sqrt{\gamma/\rho}} \frac{1}{n} \sum_{i=1}^n \xi_i v^\mathsf{T} \check{x}_i$$

$$\leq \frac{1}{n} (\gamma/\rho)^{1/4} \mathbb{E}_{\xi_i} \left\| \sum_{i=1}^n \xi_i \check{x}_i \right\| \leq \frac{1}{n} (\gamma/\rho)^{1/4} \sqrt{\mathbb{E}_{\xi_i} \left\| \sum_{i=1}^n \xi_i \check{x}_i \right\|^2}$$

$$= \frac{1}{n} (\gamma/\rho)^{1/4} \sqrt{\sum_{i=1}^n \check{x}_i^\mathsf{T} \check{x}_i}.$$

Therefore,

$$\text{Rad}(\mathcal{W}_\gamma) = \mathbb{E}[\text{Rad}(\mathcal{W}_\gamma, n)] \leq \frac{1}{n} (\gamma/\rho)^{1/4} \left( \sqrt{\sum_{i=1}^n \mathbb{E}_x \tilde{x}_i^\mathsf{T} \tilde{x}_i} + \sqrt{\sum_{i=1}^n \mathbb{E}_x \check{x}_i^\mathsf{T} \check{x}_i} \right)$$

$$\leq \frac{1}{\sqrt{n}} (\gamma/\rho)^{1/4} \left( \sqrt{\text{tr}\left( \left(\Sigma_X^{(M)}\right)^\dagger \Sigma_X \right)} + \text{rank}(\Sigma_X) \right).$$

The relationship between Rademacher complexity and generalization error [2] indicates Theorem 4. $\square$

## D.2 Two-layer ReLU Networks

We perform MSDA on the final layer of the two-layer ReLU networks. Therefore, the setting is the same as Appendix D.1 with covariates $\sigma(w_j^\mathsf{T} x)$. Due to the scaling of $\theta_1$ and $\theta_0$, we consider training

$\theta_1$ and $\theta_0$ on the covariates $1 \oslash \bar{M} \odot (M \odot (\sigma(Wx_i) - \bar{\sigma}_W) + (1 - M) \odot (\sigma(Wr_x) - \bar{\sigma}_W))$ where $\bar{\sigma}_W = \frac{1}{n} \sum_{i=1}^{m} \sigma(Wx_i)$. Putting GLM loss with $A(\cdot) = \frac{1}{2}(\cdot)^2$, we have

$$\tilde{L}_m^{(\text{MSDA})} = \frac{1}{m} \sum_{i=1}^{m} (f_\theta(x_i) - y_i)^2 + \frac{1}{m} \sum_{i=1}^{m} \left( \frac{1}{2} \theta^\mathsf{T} \text{Var}_\xi(\sigma(W(M \odot x_i + (1 - M) \odot r_x))\theta \right) \quad (27)$$

where $\tilde{L}_m^{(\text{MSDA})}$ denotes the approximate loss of $L_m^{(\text{MSDA})}$ and

$$\text{Var}_\xi(\tilde{x}_i) = \frac{1}{\lambda^2} \left( \mathbb{E}(1 - M)\hat{\Sigma}_X^\sigma (1 - M)^\mathsf{T} + \sigma(Wx_i)\text{Var}(M)\sigma(Wx_i)^\mathsf{T} \right)$$

where $\hat{\Sigma}_X^\sigma = \text{Var}_{r_x \sim \mathcal{D}_X} \sigma(Wr_x)$ with some notational ambiguity that $\frac{1}{\vec{M}} := \vec{1} \oslash \bar{M}$. Now we think of the related dual problem to the equation 27:

$$\mathcal{W}_\gamma^{\text{NN}} = \Bigg\{ x \to f_\theta(x) = \theta_1^\mathsf{T}\sigma(Wx) + \theta_0, \text{ such that } \theta \text{ satisfying}$$

$$\theta_1^\mathsf{T} \left( \mathbb{E}(1 - M)\Sigma_X^\sigma (1 - M)^\mathsf{T} \right) \theta_1 + \theta_1^\mathsf{T} \left( \mathbb{E}_x \left( \sigma(Wx)\text{Var}(M)\sigma(Wx)^\mathsf{T} \right) \right) \theta_1 \leq \gamma \Bigg\},$$

where $\Sigma_X^\sigma = \text{Var}_x \sigma(Wx)$.

**Theorem 4-(b)** (Restated). *Define* $\Sigma_X^{\sigma,(M)} = \mathbb{E}(1 - M)\Sigma_X^\sigma (1 - M)^\mathsf{T}$. *Suppose* $\mathcal{X}, \mathcal{Y}, \Theta$ *are all bounded. There exists constants* $L, B > 0$, *such that for all* $\theta$ *that* $f_\theta(x) \in \mathcal{W}_\gamma^{\text{NN}}$, *which is the regularization induced by Manifold MSDA, we have*

$$L(\theta) \leq L_m(\theta) + 4L\sqrt{\frac{\gamma \left( rank(\Sigma_X^{\sigma,(M)}) + \left\| \left( \Sigma_X^{\sigma,(M)} \right)^{\dagger/2} \mu_\sigma \right\|^2 \right)}{n}} + B\sqrt{\frac{\log(1/\delta)}{2n}},$$

*with probability at least* $1 - \delta$.

*Proof.* Firstly, we calculate the empirical Rademacher complexity of $\mathcal{W}_\gamma^{\text{NN}}$. For the $n$ i.i.d. Rademacher random variables $\xi_1, \ldots, \xi_n$, the definition of the empirical Rademacher complexity gives

$$\text{Rad}(\mathcal{W}_\gamma^{\text{NN}}, n) = \mathbb{E}_{\xi_i} \sup_{\theta_1^\mathsf{T}\left(\mathbb{E}(1-M)\Sigma_X^\sigma(1-M)^\mathsf{T}\right)\theta_1 + \theta_1^\mathsf{T}\left(\mathbb{E}_x(\sigma(Wx)\text{Var}(M)\sigma(Wx)^\mathsf{T})\right)\theta_1 \leq \gamma} \frac{1}{n} \sum_{i=1}^{n} \xi_i \theta_1^\mathsf{T}\sigma(Wx_i)$$

$$\leq \mathbb{E}_{\xi_i} \sup_{\theta_1^\mathsf{T}\left(\mathbb{E}(1-M)\Sigma_X^\sigma(1-M)^\mathsf{T}\right)\theta_1 \leq \gamma} \frac{1}{n} \sum_{i=1}^{n} \xi_i \theta_1^\mathsf{T}\sigma(Wx_i)$$

$$\leq \mathbb{E}_{\xi_i} \sup_{\theta_1^\mathsf{T}\left(\mathbb{E}(1-M)\Sigma_X^\sigma(1-M)^\mathsf{T}\right)\theta_1 \leq \gamma} \frac{1}{n} \sum_{i=1}^{n} \xi_i \theta_1^\mathsf{T}(\sigma(Wx_i) - \mu_\sigma) + \mathbb{E}_{\xi_i} \sup_{\theta_1^\mathsf{T}\left(\mathbb{E}(1-M)\Sigma_X^\sigma(1-M)^\mathsf{T}\right)\theta_1 \leq \gamma} \frac{1}{n} \sum_{i=1}^{n} \xi_i \theta_1^\mathsf{T}\mu_\sigma,$$

where $\mu_\sigma = \mathbb{E}[\sigma(Wx)]$. Setting $\tilde{\theta}_1^\mathsf{T} = \left( \Sigma_X^{\sigma,(M)} \right)^{1/2}$, same technique with the proof of Theorem 3-(a) gives

$$\text{Rad}(\mathcal{W}_\gamma^{\text{NN}}) \leq 2\sqrt{\frac{\gamma \left( rank(\Sigma_X^{\sigma,(M)}) + \left\| \left( \Sigma_X^{\sigma,(M)} \right)^{\dagger/2} \mu_\sigma \right\|^2 \right)}{n}}.$$

Finally, the relationship between Rademacher complexity and generalization error [2] indicates Theorem 4. $\qquad \square$

# E   Extending Chidambaram *et al.* [12]

Chidambaram *et al.* [12] gave the theoretical analysis of Mixup. We follow this paper by using the unified framework that we used. By modifying several definitions, we get similar results with Chidambaram *et al.* [12].

Here, we assume that $k$ classes have disjoint support, *i.e.* $X = \bigcup_{i=1}^{k} X_i$ and $X_i$ are mutually disjoint for $i = 1, \ldots, k$, where $X_i$ is the support of the $i$th class. We consider cross-entropy loss. We define an associated probability measure $\mathbb{P}_X$. Then, $L^{(\text{MSDA})}$, which is the expected loss for MSDA, can be expressed with

$$L^{(\text{MSDA})}(\theta) = \mathbb{E}_{z_1, z_2 \sim X} \mathbb{E}_{\lambda \sim \mathcal{D}_\lambda} \mathbb{E}_M l(\theta, \tilde{z}_{z_1, z_2}^{(\text{MSDA})}(\lambda, 1 - \lambda)),$$

where $l$ is the cross entropy function. We can express $L^{(\text{MSDA})}(\theta) = \sum_{i=1}^{k} \sum_{j=1}^{k} L_{i,j}^{(\text{MSDA})}(\theta)$ with $i, j \in [k]$, where $L_{i,j}^{(\text{MSDA})}(\theta)$ is defined as

$$L_{i,j}^{(\text{MSDA})}(\theta) = \mathbb{E}_{z_1, z_2 \sim X} \mathbb{E}_{\lambda \sim \mathcal{D}_\lambda} \mathbb{E}_M \left[ l(\theta, \tilde{z}_{z_1, z_2}^{(\text{MSDA})}(\lambda, 1 - \lambda)) I(z_1 \in X_i, z_2 \in X_j) \right].$$

$L_{i,j}^{(\text{MSDA})}(\theta)$ is the full MSDA cross entropy loss corresponding to the mixing points from classes $i$ and $j$. The goal of standard training is to learn a classifier $h \in \arg\min_{g \in \mathcal{C}} L(g, \mathbb{P}_X)$ where $\mathcal{C}$ is the classifier set. Any such classifier $h$ will satisfy $h(x)_i = 1$ on $X_i$ since the $X_i$ are disjoint.

We modify some definitions in [12, Section 2.2]. We define $A_{x,\epsilon}^{i,j}$ and $A_{x,\epsilon,\delta}^{i,j}$ as

$$A_{x,\epsilon}^{i,j} = \{(s, t, \lambda, M) \in X_i \times X_j \times [0,1] \times \mathbb{R}^n : M \odot s + (1 - M) \odot t \in B_\epsilon(x)\}$$

$$A_{x,\epsilon,\delta}^{i,j} = \{(s, t, \lambda, M) \in X_i \times X_j \times [0, 1 - \delta] \times \mathbb{R}^n : M \odot s + (1 - M) \odot t \in B_\epsilon(x)\}$$

$$X_{\text{MSDA}} = \left\{ x \in \mathbb{R}^n : \bigcup_{i,j} A_{x,\epsilon}^{i,j} \text{ has positive measure for every } \epsilon > 0 \right\}$$

$$\xi_{x,\epsilon}^{i,j} = \mathbb{E}_{z_1, z_2 \sim X} \mathbb{E}_{\lambda \sim \mathcal{D}_\lambda} \mathbb{E}_M [I(z_1 \in X_i, z_2 \in X_j)]$$

$$\xi_{x,\epsilon,\lambda}^{i,j} = \mathbb{E}_{z_1, z_2 \sim X} \mathbb{E}_{\lambda \sim \mathcal{D}_\lambda} \mathbb{E}_M [\lambda I(z_1 \in X_i, z_2 \in X_j)].$$

**Definition E.1** ([12]). Let $\mathcal{C}^*$ to be the subset of $\mathcal{C}$ for which every $h \in \mathcal{C}^*$ satisfies $h(x) = \lim_{\epsilon \to 0} \arg\min_{\theta \in [0,1]} L^{(\text{MSDA})}(\theta)|_{B_\epsilon(x)}$ for all $x \in X_{\text{MSDA}}$ when the limit exists. Here, $L^{(\text{MSDA})}(\theta)|_{B_\epsilon(x)}$ represents the MSDA loss for a constant function with value $\theta$ with the restriction of each term in $L^{(\text{MSDA})}$ to the set $A_{x,\epsilon}^{i,j}$

$\mathcal{C}^*$ includes deep neural networks. Below lemmas and theorems can be proved with the same technique as Chidambaram *et al.*.

**Lemma 1.** *Any function $h \in \arg\min_{g \in \mathcal{C}^*} L^{(MSDA)}(g, \mathbb{P}_X, \mathbb{P}_f)$ satisfies $L^{(MSDA)}(h) \le L^{(MSDA)}(g)$ for any continuous $g \in \mathcal{C}$*

**Theorem 6.** *For any point $x \in X_{MSDA}$ and $\epsilon > 0$, there exists a continuous function $h_\epsilon$ satisfying*

$$h_\epsilon^i(x) = \frac{\xi_{x,\epsilon}^{i,i} + (\sum_{j \neq i} \xi_{x,\epsilon,\lambda}^{i,j} + (\xi_{x,\epsilon}^{j,i} - \xi_{x,\epsilon,\lambda}^{j,i}))}{\sum_{q=1}^{k} \xi_{x,\epsilon}^{q,q} + \sum_{j \neq q} (\xi_{x,\epsilon,\lambda}^{q,j} + (\xi_{x,\epsilon}^{j,q} - \xi_{x,\epsilon,\lambda}^{j,q}))},$$

*and its limits exist when $\epsilon \to 0$.*

We give an assumption: for a point $x \in X_{\text{MSDA}}$, there exists a class $i$ that $x$ is closest to $X_i$ for arbitrary MSDA expression of $x$, and $x$ cannot be expressed by MSDA expression between non-$i$ classes. The formal assumption and its geometric intuition can be found in [12].

**Theorem 7.** *If $x$ satisfies the above assumption, with respect to a class $i$, then for every $h \in \arg\min_{g \in \mathcal{C}^*} L_{MSDA}(g)$, we have that $h$ classifies $x$ as the class $i$ and $h$ is continuous at $x$.*

Theorem 7 indicates that if we observe a new sample that can satisfy the above assumption with the class $i$, which is also a distance up to $\min_j d(X_i, X_j)/2$ from the class $i$, the model trained with MSDA will classify it as $i$. Therefore, Theorem 7 is closely related to the generalization properties of MSDA. All proof of this section is identical to Chidambaram *et al.*.

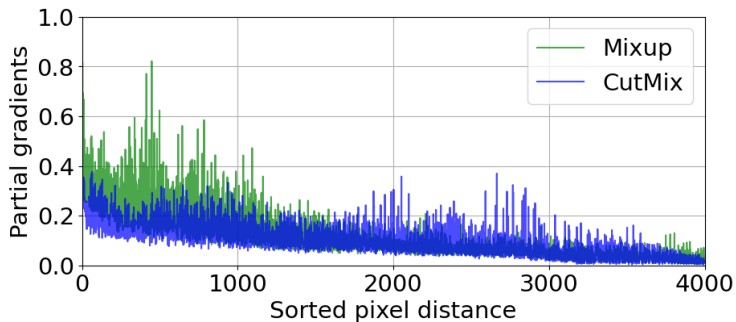

Figure 6: Comparison of regularized partial gradients between Mixup and CutMix along the sorted pixel indices.

Table 4: **HMix performances in both scenarios of Table 1.**

|  | Mixup | CutMix | Δ (CutMix - Mixup) | HMix ($r$=0.5) | HMix ($r$=0.75) |
|---|---|---|---|---|---|
| Scenario 1: Large crop | 58.3 | **64.4** | **+6.1** | **61.3** (**+3.0** vs. Mixup) | **63.7** (**+5.4** vs. Mixup) |
| Scenario 2: Small crop | **67.7** | 67.0 | **-0.7** | **67.6** (**+0.6** vs. CutMix) | **67.2** (**+0.2** vs. CutMix) |

# F    Experimental Details

**Regularized input gradients experiments (Figure 4).**    We investigate the amount of the regularized input gradients by $|\partial_v f_\theta(x)\partial_{v+p} f_\theta(x)|$ with respect to the pixel distance vector $p$. We then compute the partial gradients for the validation images $x$, which have not been seen during training, and normalize them to sum to 1 as

$$\text{PartialGradProd}(x, p) = \max_v |\partial_v f_\theta(x)\partial_{v+p} f_\theta(x)|$$

for different $f_\theta$ trained by different MSDA methods. Finally, we visualize the pixel-wise maximum values of PartialGradProd$(x, p)$ for the validation images $x$ in Figure 4. We train ResNet-50 [26] models on the ImageNet-1K dataset with $64 \times 64$ image size. We show additional visualization in Figure 6, where the $x$-axis denotes the pixel indices sorted by the pixel distance $\|p\|$. As shown in Figure 4 and Figure 6, CutMix more regularizes the partial gradient products $|\partial_v f_\theta(x)\partial_{v+p} f_\theta(x)|$ for a closer $p$ than Mixup.

**CIFAR-100 experiments.**    We follow the experimental setting of Kim *et al.* [39] and their official implementation[2]. We train all networks for 300 epochs using the SGD optimizer with a learning rate of 0.2 and batch size of 100. The learning rate is decayed by a factor of 0.1 at 100 and 200 epochs. We set the hyper-parameter $\alpha$ for Mixup, CutMix, and Stochastic Mixup & CutMix to 1. $\alpha$ for HMix and GMix were set to 1 and 0.5, respectively. We use $r = 0.5$ for HMix. All the experiments in CIFAR-100 were repeated three times, and we report the average accuracy.

**ImageNet-1K experiments.**    We use *timm*[69][3] library for our implementation. We train ResNet-50 [26] for 300 epochs using the SGD optimizer with a learning rate of 0.1, weight decay of $2 \times 10^{-5}$, and batch size of 512. We use the cosine learning rate scheduling. We set the hyper-parameter $\alpha$ to 1, except Mixup; Mixup has $\alpha = 0.8$. We use $r = 0.75$ for HMix.

# G    Additional Experiments

**HMix results for the scenarios of Table 1.**    Table 4 shows the HMix results on two scenarios of Table 1. We use $r = 0.5$ and $r = 0.75$ for HMix. HMix shows consistently better performance than Mixup and CutMix for scenario 1 and 2, respectively, with meaningful performance gaps. From the results, we empirically confirm HMix enjoyed the advantages of Mixup and CutMix.

---

[2]https://github.com/snu-mllab/PuzzleMix
[3]https://github.com/rwightman/pytorch-image-models

Table 5: **Robustness benchmarks.** Comparison of various MSDA methods on ResNet-50 architecture.

| Augmentation Method | ImageNet-1K | ImageNet-occ | ImageNet-C | FGSM |
|---|---|---|---|---|
| Vanilla (no MDSA) | 75.68 (+0.00) | 55.26 (+0.00) | 42.57 (+0.00) | 8.55 (+0.00) |
| Mixup | 77.78 (+2.10) | 60.34 (+5.08) | **51.73 (+9.16)** | 27.78 (+19.23) |
| CutMix | 78.04 (+2.36) | **71.51 (+16.25)** | 44.18 (+1.55) | 33.63 (+25.08) |
| **HMix** (ours) | **78.38** (+2.70) | 71.13 (+15.87) | 46.37 (+3.80) | **34.98 (+26.44)** |
| **GMix** (ours) | 78.13 (+2.45) | 62.76 (+7.50) | 45.97 (+3.40) | 31.02 (+21.47) |

Table 6: **Ablation study on hyper-parameters.**

(a) **Impact on $r$ for HMix.**

| PreActRN18 | CIFAR-100 acc |
|---|---|
| Vanilla (no MDSA) | 76.73 |
| Mixup ($r = 0$) | 77.21 |
| CutMix ($r = 1.0$) | 78.66 |
| **HMix** ($r = 0.75$) | **79.43** |
| **HMix** ($r = 0.5$) | 79.25 |
| **HMix** ($r = 0.25$) | 78.05 |

(b) **Impact on $\alpha$ for HMix.**

| PreActRN18 | CIFAR-100 acc |
|---|---|
| Vanilla (no MDSA) | 76.73 |
| Mixup ($\alpha = 1.0$) | 77.21 |
| CutMix ($\alpha = 1.0$) | 78.66 |
| **HMix** ($\alpha = 0.5$) | 78.47 |
| **HMix** ($\alpha = 1.0$) | **79.25** |
| **HMix** ($\alpha = 2.0$) | 78.85 |

(c) **Impact on $\alpha$ for GMix**

| PreActRN18 | CIFAR-100 acc |
|---|---|
| Vanilla (no MDSA) | 76.73 |
| Mixup ($\alpha = 1.0$) | 77.21 |
| CutMix ($\alpha = 1.0$) | 78.66 |
| **GMix** ($\alpha = 0.25$) | 78.60 |
| **GMix** ($\alpha = 0.5$) | **79.17** |
| **GMix** ($\alpha = 0.75$) | 78.64 |
| **GMix** ($\alpha = 1.0$) | 79.05 |

**Robustness benchmarks.** As observed by the previous study [13], the different choice of MSDA methods also affects the extreme case of the test samples, *e.g.*, distribution shifts. In this experiments, we provide an understanding of the relationship between MSDA and various test scenarios and through the lens of our theoretical analysis. We conduct various MSDA methods on robustness benchmarks such as ImageNet-1K occlusion accuracy (center occluded images following [72, 13]), ImageNet-C accuracy [28] and adversarially attacked ImageNet test accuracy by FGSM attack [22]. Results are in Table 5. We use the same experimental settings of the ImageNet classification in Table 3. Overall results show that HMix and GMix are located in between CutMix and Mixup.

CutMix performs better than Mixup in the occlusion accuracy. Here, the local occluded areas have no information to distinguish objects, but other local areas are informative. Hence, it is important to capture shorter-relationship rather than global-relationship. Thus we can expect that CutMix is better than Mixup, and not surprisingly, HMix and GMix are located in between CutMix and Mixup.

ImageNet-C style corruptions (*e.g.*, adding Gaussian noise for the entire image) distort the local information. In this case, the shorter-distance relationships are significantly damaged and sometimes useless to distinguish the object, hence the longer-distance relationships are more important. Hence, we can expect Mixup works better than CutMix in ImageNet-C. As HMix and GMix less weight shorter-distance relationships than CutMix (but more weight than Mixup), we can observe that HMix and GMix ImageNet-C performances are better than CutMix, but worse than Mixup.

**Ablation study on hyper-parameters.** Table 6 shows the impacts of hyper-parameters $\alpha$ and $r$ for HMix and GMix on CIFAR-100 classification. We use the PreActResNet-18 with the same experimental setting as in Table 2. We highlight the results with our hyper-parameter choices ($r = 0.5$, $\alpha = 1.0$ for HMix, $\alpha = 0.5$ for GMix) in the gray cells. For $r$, we find that HMix with $r = 0.75$ performs better than $r = 0.5$ with a marginal gap (+0.18%p). For $\alpha$, our hyper-parameters show the best performance. Overall results confirm that HMix and GMix are not sensitive to those hyper-parameters and consistently show better or compatible performance against Mixup and CutMix.

# H  Regularizer coefficients of HMix

Define

$$h(x,s) = \min(x, n-s), \qquad l(x,s) = \max(x-s, 0),$$

$$a_{jk,s} = \frac{\max(\min(h(j_1,s)-l(k_1,s), h(k_1,s)-l(j_1,s)), 0) \max(\min(h(j_2,s)-l(k_2,s), h(k_2,s)-l(j_2,s)), 0)}{(n-s)^2},$$

(28)

$$o_s = \frac{\lambda n^2}{n^2 - s^2},$$

$$v(p,s) = \frac{(h(p_1,s)-l(p_1,s)) * (h(p_2,s)-l(p_2,s))}{(n-s)^2}.$$

Note that (28) is extension of (10) by putting $s = [\sqrt{1-\lambda}n]$ in (28). Then, HMix with hyperparameter $r$ has regularizer coefficient $a_{ij}$ as

$$s = [\sqrt{1-\lambda}\sqrt{r}n]$$
$$a_{ij} = o(s)(1-o(s))(v(i,s)+v(j,s)) + o(s)a_{ij,s} + (1-o(s))(1-o(s)).$$

We plotted this value in Figure 4 when $r = 0.7$.

# I  What MSDA can be applied in our Thereoms?

Our theorems can be applied to any MSDA method with an analogous formula, regardless of the assumption of the shape of the mask. In this paper, we mainly focused on Mixup and CutMix because they are the most common MSDA methods among the whole MSDA family as well as their behaviors are distinctly different in terms of our theorem. In this section, we note several nontrivial remarks for understanding the setup of our paper.

**ResizeMix.**  ResizeMix [54] can be explained by our theorem if we add the assumption on the dataset $\mathcal{D}_X$ that $\mathcal{D}_X$ has all resized versions of the image. ResizeMix uses the resized version of input (i.e., one of the mixed patches is the "resized" version, not a cropped one) where the random resize is applied to the whole dataset. In other words, ResizeMix is a special case of CutMix when we apply a special version of random resize crop operation. Hence, if we assume a different version of random resize crop rather than the standard version (independent of our theoretical results and underlying assumptions), ResizeMix is equivalent to CutMix, which leads to the same theoretical result as CutMix.

**FMix.**  FMix [25] randomly samples the mask from the Fourier space. Since FMix is one of the static MSDA, we can directly apply our Theorem 1-4.

**SaliencyMix, PuzzleMix, and Co-Mixup.**  SaliencyMix [65] uses saliency map to generate new MSDA sample. PuzzleMix [39], and Co-Mixup [38] are dynamic MSDA, where they use saliency map and transport. These methods give a state-of-the-art performance. Theorem 1 can deal with this problem, but hard to interpret. To be specific, the second equality of Equation (8) do not hold anymore; it is hard to interpret the approximated loss function as an input gradient / Hessian regularizer.

**StyleMix and Manifold Mixup.**  StyleMix [29] uses pre-trained style encoder and decoder. StyleMix linearly mixes content and style. Manifold Mixup [66] mixes samples in the feature level. Therefore, the theorems cannot be directly applied to StyleMix and Manifold Mixup.

**AutoMix.**  Recently, AutoMix [45] gives state-of-the-art results. AutoMix utilized joint loss to generate the mask $M$: classification loss and generation loss for training $M$. Therefore, the mask depends on the mixing samples, indicating that AutoMix is a dynamic MSDA. As in previous paragraphs, Theorem 1 holds, but it is not easy to interpret each term.