# OpenReview forum: "A Unified Analysis of Mixed Sample Data Augmentation: A Loss Function Perspective"
_NeurIPS.cc/2022/Conference — NeurIPS 2022 Accept_

### Official Review · Reviewer_phJK · 2022-07-10

**Rating:** 8
**Confidence:** 4
**Soundness:** 4 excellent
**Presentation:** 4 excellent
**Contribution:** 4 excellent

**Summary:**

This paper proposes unified theoretical analysis of mixed sample data augmentation (MSDA) by extending Zhang et al. (2021) that analyzed theoretical model of Mixup. The authors show that MSDSA can improve performance by regularization effects in gradient and Hessian. Furthermore, they propose new augmentation methods, HMix and GMix, based on their theoretical analysis. Experimental results show that HMix and GMix outperform the previous MSDA methods in CIFAR-100 and ImageNet.

**Questions:**

1. Is second-order Taylor approximation of loss function valid? Zhang et al. (2021) suggest that the approximation is valid for Mixup. Can their result be generalized to other MSDA?

2. Can results of Theorem 1 are applied to other MSDA methods such as ResizeMix (Qin et al., 2020), Fmix (Harris et al., 2021), or PuzzleMix?

- Qin, J., Fang, J., Zhang, Q., Liu, W., Wang, X., and Wang, X. (2020). Resizemix: Mixing data with preserved object information and true labels. arXiv preprint arXiv:2012.11101.
- Harris, E., Marcu, A., Painter, M., Niranjan, M., Prügel-Bennett, A., and Hare, J. Fmix: Enhancing mixed sample data augmentation. ICLR, 2021.


**Limitations:**

This paper does not consider online optimization of mask design $M$ or $N$. It may be an interested topic for future works.

**Strengths And Weaknesses:**

Strengths:

- This is a first theoretical study that proposes unified theoretical analysis of MSDA. The authors show theoretical reason why MSDA works. Furthermore, they show that Mixup and CutMix have different regularization effects in gradient and Hessian.

- The proposed HMix and GMix are consistent with their theoretical analysis. HMix and GMix outperform the previous MSDA methods in CIFAR-100 and ImageNet.

- The paper is clearly written.

Weaknesses:

- Theorem 1 requires second-order Taylor approximation. This paper does not consider the validity of this approximation.

- The authors do not provide $\alpha_{ij}$ of HMix, while they provide the one of Mixup, CutMix, and GMix.

- The authors argue that potential negative societal impacts are discussed in the appendix. However, I cannot find the discussion in the appendix.

---

> ### Author Response · Authors · 2022-08-02
> **Response to Reviewer phJK**
>
> We thank Reviewer phJK for positive comments and constructive suggestions for improving our work. We are also pleased to hear that the reviewer found (1) our theoretical result is the first study to explain why MSDA works (2) our theoretical results can show how Mixup and CutMix are different in terms of gradient and Hessian regularization (3) our theoretical analysis is consist with the proposed HMix and GMix (4) our paper is well-written.
>
> We addressed all concerns raised by the reviewer and revised our paper accordingly.
>
>
> ### No validation of second-order Taylor approximation (W1, Q1)
> Thanks for the constructive feedback. We also agree that our theorem can be better if we can validate the second-order Taylor approximation. Hence, we validate the second-order Taylor approximation following Zhang et al. [R1]. We employ the same toy dataset of Zhang et al. and employ two simple MSDA methods: (1) randomly and independently chooses its mask either 0 or 1 for each input dimension (2) a simple Mixup. We visualize the approximated loss functions of the employed MSDA methods and the true MSDA loss functions. We add the figures in Figure 1 of the revised version of paper. In the figure, we can observe that the approximation gaps between the true loss functions and the approximated loss functions are small as also observed by Zhang et al.
>
> ### Extension to other MSDA methods (ResizeMix, FMix, PuzzleMix)?
> Our theorems can be applied to any MSDA method with an analogous formula, regardless of the assumption of the shape of the mask. In this paper, we mainly focused on Mixup and CutMix because they are the most common MSDA methods among the whole MSDA family as well as their behaviors are distinctly different in terms of our theorem.
>
> <ResizeMix>
>
> ResizeMix can be explained by our theorem if we add the assumption on the dataset $\mathcal{D}_X$ that $\mathcal{D}_X$ has all resized versions of the image. ResizeMix uses the resized version of input (i.e., one of the mixed patch is the “resized” version, not a cropped one) where the random resize is applied to the whole dataset. In other words, ResizeMix is a special case of CutMix when we apply a special version of random resize crop operation. Hence, if we assume a different version of random resize crop rather than the standard version (which is independent to our theoretical results and underlying assumptions), ResizeMix is equivalent to CutMix that leads to the same theoretical result to CutMix.
>
> <FMix>
>
> FMix randomly samples the mask from the Fourier space. Since FMix is one of the static MSDA, we can directly apply our Theorem 1-4.
>
> <PuzzleMix>
>
> PuzzleMix samples the mask depending on the saliency map of the given input. Therefore, PuzzleMix is a dynamic MSDA, while we can apply our Theorem 1, but it is not straightforward to interpret what is the meaning of regularizer.
>
>
> ### Other comments (W3, W2)
> - No $\alpha_{ij}$ of HMix (W2): Thanks for the comment. We added it in the Appendix.
> - No negative societal impacts: Sorry for the mistake. We added the negative societal impacts in the revised Appendix.
>
> [R1] Zhang, Linjun, et al. "How does mixup help with robustness and generalization?." ICLR 2021

---

> > ### Comment · Reviewer_phJK · 2022-08-09
> > **Response to Authors**
> >
> > Thank you for the rebuttal.
> > The overall results look good to me.

---

### Official Review · Reviewer_h7Ks · 2022-07-11

**Rating:** 5
**Confidence:** 5
**Soundness:** 2 fair
**Presentation:** 3 good
**Contribution:** 2 fair

**Summary:**

This paper proposes a unified theoretical analysis of mixed sample data augmentation (MSDA). The theory shows that the MSDA training strategy can improve adversarial robustness and generalization of training. Based on these results, this paper proposes generalized MSDAs, a Hybrid version of Mixup and CutMix (HMix) and Gaussian Mixup (GMix), which are simple extensions of Mixup and CutMix. Where HMix unifies Mixup and Cutmix and uses the Beta distribution to determine the hybrid strategy hyperparameters, GMix is a smoother hybrid strategy.

**Questions:**

- The advantage of the beta distribution in HMix is not explained, simply because the Beta distribution is more commonly used and the Beta distribution has a closed-form of conjugate representation for easy derivation?
- The proposed strategies do not consider the label mismatch problem [4]; they both construct hybrid strategies by randomly selecting a box or a point. If the mixed samples and labels do not match, does it affect the theoretical derivation of conclusions?
- Both proposed strategies do not take full advantage of the previous conclusions, for example, MSDA is data-dependent, but both hybrid strategies are data-independent.
- The effect of hyperparameters such as \alpha, r, \lambda, etc. in the hybrid strategy is not fully explored.

[4] Liu, Zicheng, et al. "Automix: Unveiling the power of mixup." European Conference on Computer Vision, 2022.

**Limitations:**

Based on previous theoretical works on mixup methods, this paper draws some interesting conclusions. However, the proposed method is very similar to the already published SmoothMix and the methods compared in the experimental section are not adequate. Therefore, I think this article is not up to the acceptance criteria.

**Strengths And Weaknesses:**

### Strengths
- The paper is well written and organized.
-  Detailed theoretical derivations support the ideas of the paper.

### Weaknesses
- Although the paper draws some conclusions through theoretical derivation, the proposed method is too similar to SmoothMix [1].
- Some classical baselines, such as ManifoldMix [2], are missing from the experimental comparison methods. in addition, some new mixup methods of comparison are missing, such as SaliencyMix [3] (comparable in computational complexity to vanilla mixup), etc.
- Some problems of theoretical derivation (see the questions for details).

[1] Lee, Jin-Ha, et al. "Smoothmix: a simple yet effective data augmentation to train robust classifiers." Proceedings of the IEEE/CVF conference on computer vision and pattern recognition workshops, 2020.

[2] Verma, Vikas, et al. "Manifold mixup: Better representations by interpolating hidden states." International Conference on Machine Learning. PMLR, 2019.

[3] Uddin, A. F. M., et al. "Saliencymix: A saliency guided data augmentation strategy for better regularization." International Conference on Learning Representations, 2021.

---

> ### Author Response · Authors · 2022-08-02
> **Response to Reviewer h7Ks (1)**
>
> We thank the reviewer for their time and suggestions for improving our submission. We are happy to hear that the reviewer found the paper well-organized.
>
> The main goal of our paper is to understand various MSDAs from the optimization perspective and to provide a generalized loss function form. We also have shown that there is no one-fit-all solution for MSDA, resulting in proposing a hybrid version of popular MSDA methods, i.e., Mixup and CutMix. We address the reviewer’s comments below.
>
>
> ### Comparison with SmoothMix (W1 and Limitation 1)
> Our paper focuses on theoretical contribution (a unified theoretical framework for general MDSA). HMix, GMix are our new methods, but we propose the methods for supporting our theoretical results (as the reviewer also agreed), rather than proposing a novel data augmentation method. On the other hand, SmoothMix [R1] focuses on alleviating the strong-edge effect; while our HMix and GMix focus on leveraging the advantages of both CutMix and Mixup with a theoretical understanding. Even though our method can look similar to SmoothMix, because the main motivation of the papers and the conclusion are completely different, we argue that SmoothMix does not weaken our contribution, such as unified theoretical analysis of MSDA and empirical analyses for supporting our main theorem.
>
> We also point out the main differences between our proposed methods and SmoothMix below, so we claim that the proposed algorithms and SmoothMix are entirely different.
> - SmoothMix-C focuses on alleviating the strong-edge effect. Therefore, the mask value gradually decreases with rectangular shapes. However, HMix focuses on leveraging the advantages of CutMix and Mixup, so the mask of HMix does not gradually decrease. Moreover, the parameter of SmoothMix-C ($\sigma$) is uniformly selected in the original paper [R1], but we firstly select $\lambda$, which is the mix ratio of two samples in Beta distribution as Mixup and CutMix. Second, HMix can change the hybrid ratio with $r$, which also has Beta distribution,  while SmoothMix-C has a fixed value of $k$ for alleviating the strong-edge effect. We think this is a significant difference between SmoothMix-C and HMix since HMix can recover both Mixup and CutMix by selecting the parameter of the Beta distribution. As we mentioned in our paper, the performance of MSDA depends on the specific task or domain, and our theorems also support it, so we should consider diverse situations and use several candidates between Mixup and Cutmix. Therefore, while SmoothMix-C is also a great idea, intuition is completely different.
> - SmoothMix-S and GMix also differ in parameter selection. The parameter of SmoothMix-S ($\sigma$) is also selected from the uniform distribution, while GMix selects parameters from Beta distribution as Mixup and CutMix.
> We think this parameter selection issue is important since we directly control the mixing ratio distribution with Beta distribution, but in SmoothMix, more focuses on alleviating the strong-edge effect, which can not directly control the distribution of the mixing ratio. Controlling the mixing ratio is also an empirically important issue, as alleviating the strong-edge effect. Therefore, these two algorithms are different.
>
> ### Not enough comparison methods (W2 and Limitation 2)
> We add two additional comparison methods (ManifoldMixup and SaliencyMix). Every experiment is done with the same hyperparameter from each paper.
>
> 1. Our paper mainly focuses on pixel-level MSDA (pixel-level mix operations), while ManifoldMixup [R2] is a feature-level MSDA. Note that, we also have shown that feature-level MSDA also guarantees the generalization bounds in Theorem 4. We additionally report the ManifoldMixup result.
> 2. We also mainly focus on data independent MSDA (e.g., Mixup and CutMix), while SaliencyMix [R3] is a data dependent method. Note that our main Theorems also hold for a data dependent MSDA, but we do not analyze data dependent MSDA methods because the meaning of loss function is not clear for data dependent MSDA; because the second equality of (7) and (8) do not hold anymore, it is hard to interpret the approximated loss function as an input gradient / Hessian regularizer. We show the SaliencyMix results in the below table.
>
> | PreActRN18     | CIFAR-100 accuracy |
> |----------------|--------------------|
> | Mixup          |              77.21 |
> | CutMix         |              78.66 |
> | SaliencyMix    |              77.84 |
> | Manifold Mixup |              78.63 |
> | HMix           |          **79.25** |
> | GMix           |          **79.17** |

---

> > ### Comment · Reviewer_h7Ks · 2022-08-07
> > **Response to Authors**
> >
> > Thank you for the rebuttal.
> > The overall results look good to me now.
> > However, the authors have missed citing several state-of-the-art mixup works such as Co-Mixup [1], SaliencyMix [2], StyleMix [3], AutoMix [4], etc. It would be nice to discuss these papers in the related work section and I will raise my score. It is best to compare them if possible.
> >
> > [1] Kim et al., Co-mixup: Saliency guided joint mixup with supermodular diversity. ICLR 2021.
> >
> > [2] Uddin et al. Saliencymix: A saliency guided data augmentation strategy for better regularization. ICML 2021.
> >
> > [3] Hong et al., Stylemix: Separating content and style for enhanced data augmentation. CVPR, 2021.
> >
> > [4] Liu, et al. AutoMix: Unveiling the Power of Mixup for Stronger Classifiers. ECCV, 2022.

---

> > > ### Author Response · Authors · 2022-08-09
> > > **Thank you**
> > >
> > > We are happy to hear that our revision version of the paper clarifies our contribution. Thanks for the recommendation of the state-of-the-art MSDA papers. We note that we already mentioned [1,2, 4-8] in the original paper. We added the discussion in Appendix J that dynamic MSDA could or not be applied to our theorems, including [1-8].
> > >
> > > [1] Kim et al., Co-mixup: Saliency guided joint mixup with supermodular diversity. ICLR 2021.
> > >
> > > [2] Uddin et al. Saliencymix: A saliency guided data augmentation strategy for better regularization. ICML 2021.
> > >
> > > [3] Hong et al., Stylemix: Separating content and style for enhanced data augmentation. CVPR, 2021.
> > >
> > > [4] Liu, et al. AutoMix: Unveiling the Power of Mixup for Stronger Classifiers. ECCV, 2022.
> > >
> > > [5] Kim et al. Puzzle mix: Exploiting saliency and local statistics for optimal mixup. ICML, 2020
> > >
> > > [6] Verma, et al. Manifold mixup: Better representations by interpolating hidden states. ICML 2019
> > >
> > > [7] Qin et al., Resizemix: Mixing data with preserved object information and true labels. arXiv preprint arXiv:2012.11101.
> > >
> > > [8] Harris et al., Fmix: Enhancing mixed sample data augmentation. ICLR, 2021.

---

> ### Author Response · Authors · 2022-08-02
> **Response to Reviewer h7Ks (2)**
>
>
> ### HMix and GMix do not consider label mismatch problem
> We first emphasize that our theoretical results are invariant to the existence of the label mismatch problem; there is no assumption on data points, there is no assumption on mask. Moreover, **our MSDA loss function formulation depends on the distribution of the mask, not the individual mask sampling** for each data augmentation step. For example, one can imagine an extreme case of label mismatch such as mixing two backgrounds without any object. However, our mask distribution also allows other cases, such as two images mixed well with proper labels. In other words, some individual mask sampling can suffer from extreme label mismatch problems, but it is not a problem of mask distribution itself.
>
> More specifically, our approximated loss function (6) always holds although some individual mask sampling suffers from label mismatch problems. Moreover, as shown in our theorem, the regularization terms (8) are only determined by the formulation of the mask distribution: $a_{jk} := E_M[(1-M_j)(1-M_k)]$ (9). If we sample masks in a data independent manner, then **$a_{jk}$ is independent of the data points, which means that the label mismatch problem does not change our theoretical derivation**. Therefore, our proposed HMix and GMix do not consider label mismatch problems, and it does not hurt our theoretical results, because label mismatch problem is only dependent on the individual mask sampling.
>
>
> ### Both hybrid strategies are data-independent
> In this paper, we focus on explaining data independent MSDA methods, i.e., $M$ is a random variable only depending on $\lambda$, but independent to $x$ (L80-81 and Remark 2). As we already emphasized that our proof techniques also can be applied to the dynamic MSDA methods because our theoretical analysis are invariant to the choice of $M$ and $N$. We agree that data dependent MSDA could be an interesting research topic, but our study is independent of data dependent mask selection strategy and any data independent mask selection strategy also can enjoy our theoretical results.
>
> ### Hyperparameter analysis
> We thank the reviewer for pointing out the issue that the effect of hyperparameter is not sufficiently discussed. We added the effect of hyperparameters in this rebuttal comment and the revised version of the manuscript.
>
> <Impact on $r$ for HMix>
> | PreActRN18      | CIFAR-100 accuracy |
> |-----------------|--------------------|
> | Mixup           |              77.21 |
> | CutMix          |              78.66 |
> | HMix ($r$=0.75) |          **79.43** |
> | HMix ($r$=0.5)  |              79.25 |
> | HMix ($r$=0.25) |              78.05 |
>
> <Impact on $\alpha$ for HMix>
> | PreActRN18                      | CIFAR-100 accuracy |
> |---------------------------------|--------------------|
> | Mixup                           |              77.21 |
> | CutMix                          |              78.66 |
> | HMix ($\alpha$=0.5, $r$=0.5) 	   |              78.85 |
> | HMix ($\alpha$=1.0, $r$=0.5)    |          **79.25** |
> | HMix ($\alpha$=2.0, $r$=0.5)    |              78.47 |
>
> <Impact on $\alpha$ for GMix>
> | PreActRN18           | CIFAR-100 accuracy |
> |----------------------|--------------------|
> | Mixup                |              77.21 |
> | CutMix               |              78.66 |
> | GMix ($\alpha$=0.25) |              78.60 |
> | GMix ($\alpha$=0.5)  |          **79.17** |
> | GMix ($\alpha$=0.75) |              78.64 |
> | GMix ($\alpha$=1)    |              	79.05 |
>
>
> [R1] Lee, Jin-Ha, et al. "Smoothmix: a simple yet effective data augmentation to train robust classifiers." Proceedings of the IEEE/CVF conference on computer vision and pattern recognition workshops, 2020.
>
> [R2] Verma, Vikas, et al. "Manifold mixup: Better representations by interpolating hidden states." International Conference on Machine Learning. PMLR, 2019.
>
> [R3] Uddin, A. F. M., et al. "Saliencymix: A saliency guided data augmentation strategy for better regularization." International Conference on Learning Representations, 2021.

---

### Official Review · Reviewer_4bQe · 2022-07-13

**Rating:** 7
**Confidence:** 3
**Soundness:** 3 good
**Presentation:** 3 good
**Contribution:** 3 good

**Summary:**

The authors develop and unified theory to explain the various data augmentation (DA) schemes used in Deep network training literature. Particularly, taking motivation from mixed sample strategies (MSDA) and cropping based approaches such as CutMix, the authors propose a loss formulation that considers any non-DA loss, and can be used to generalize various masking and mixing approaches to DA. They show that the CutMix and MSDA losses can then be interpreted through first- and second-order gradient regularization, thereby providing a framework to incorporate/develop other DA schemes. They propose two alternatives DA schemes, HMix and Mix, that result in improved performance of certain classification tasks.

While I did not explore these results completely, the inclusion of results on n-sample DA in the Supplementary helps broaden the scope of the future work/applications of the proposed results in the paper.

**Questions:**

While I understand the space constraints present, some points in the Theorem’s proof could still be discussed in the Main paper, to help better understand the results — I’m not sure if this a standard assumption in the field, but it seemed unclear to me why one would need $E_{r_x}[r_x] = 0$, or if there are any specific practical implications to having this assumption. Could we guarantee this on the datasets?



**Limitations:**

I found the paper to be self contained and the authors seems to have sufficiently discussed most topics that would appear to be open ends for future work, such as explore scenarios with negative M in MSDA, etc.

**Strengths And Weaknesses:**

 - While my expertise in DA is not specific to loss formulation and analysis topics covered in the paper, the results presented in the paper help improve the overall understanding of how various DA schemes affect the learning objective, which is clearly presented.
 - To the best of my knowledge, the proposed general loss function proposed is a significant contribution and should help the community analyze other DA schemes as well, those not considered in the paper.
 - The empirical results on HMix and GMix help validate the formulation developed.
 - The paper is generally well written, with a clear flow of thoughts and presentation, and is clear to read even for those with minimal expertise in the particular sub-field that the papers is about.
 - Minor Issue: The paper could use a bit of proofreading to fix typos here and there, eg. L147 can be benefit -> can be beneficial.

==== x ==== x ==== x ====

Post-Rebuttal comments: I have gone though the author's responses to my and the other reviewers' comments and as there is an overall consensus that the paper has good merit and is deserving of an acceptance, I will keep my score to accept the paper.

---

> ### Author Response · Authors · 2022-08-02
> **Response to Reviewer 4bQe**
>
> We thank the reviewer for the constructive and positive reviews. We are happy to hear that Reviewer 4bQe agree with that (1) our generalized loss formulation and analysis help overall understanding of how MDSA scheme works (2) our general loss function is a significant contribution and should be helpful for other MSDA methods (3) HMix, GMix results support our claim (4) our paper is well written and clear.
>
> We answer all questions raised by the reviewer and revise the manuscript accordingly.
>
>
> ### Q1. Can the proof be self-contained?
> We agree with the reviewer. We revised our main manuscript to self-contain the main proof sketch, for example:
> - Proof sketch for Theorem 1: Using the definition of $z_{ij}$ and using the fact that the Binomial distribution and Beta distribution are in the conjugate, we can reformulate $L_m^{(MSDA)}$. In the process of reformulating $L_m^{(MSDA)}$, we should define $D_\lambda$. Then, we can make a quadratic Taylor approximation of the loss term. Here, $E_{r_x}[r_x]=0$ is used for not only the simplicity of the results, but also for the fact that using normalization in the dataset. Details can be found in Appendix.
> - Proof sketch for Theorem 3: Defining adversarial loss function and using second order taylor expansion, we can prove that adversarial loss is less than MSDA loss.
> - Proof sketch for Theorem 4: MSDA regularization can be altered to the original empirical risk minimization problem with a constrained function set, and calculating Radamacher complexity of this function set gives the theorem.
> The revised explanations are in Section 3 of the revised Supplementary Material.
>
>
> ### Q2. Why we need $E_{r_x}[r_x] = 0$ assumption?
> We assume $E_{r_x}[r_x] = 0$ condition (i.e., the mean of the given data points is 0) due to the simplification of the theoretical results. With a simple modification of the proof, we can get the almost same result but the coordination is parallelly shifted by the mean of the dataset. In practice, one can easily make a 0-centered dataset by simply moving all data points that have a mean of 0. Note that we actually make a zero-centered dataset for training deep neural networks by subtracting the mean of the dataset to a stable training (e.g., `Normalize(mean, std)` operation in torchvision package).
>
> ### Minor issues:
> - We fixed typos in the revised paper.
> - Thanks for enjoying our paper regarding n-sample DA. We included a more detailed description about  n-sample DA in the main manuscript (revised Supplementary Material L147-L149) for better understanding. Thanks for the comment.

---

### Official Review · Reviewer_ynng · 2022-07-14

**Rating:** 7
**Confidence:** 2
**Soundness:** 3 good
**Presentation:** 3 good
**Contribution:** 3 good

**Summary:**

This paper extends the previous theoretical analysis on Mixup to all Mixup-based methods with data-agnostic mask selection methods. Inspired by the previous analysis on Mixup, this work also shows that Mixup-based methods help improve generalization and robustness performances. And this paper shows that Mixup-based methods can be viewed as the regularizer of input gradient and Hessian as well as the first layer parameters. From such a view, this work further investigates that different Mixup-based methods have different effects as the regularizer on input gradient and Hessian and hence there is no optimal choice for all the cases. Stand on that, this paper proposed two combinations of Mixup and CutMix that combine the advantages of both of them.

**Questions:**

1. I didn't find Theorem 4 in the Appendix. Do Theorem 3 (a) and (b) in the Appendix correspond to Theorem 4?

2. For understanding Fig. 4, could you add the meanings of the x & y-axes and the color bar? Are the x & y-axes mean the distances between pixels along a specific dimension instead of the location of a pixel?


**Limitations:**

1. This paper claims that Mixup & CutMix are good for different cases. But it only provides the results of their proposed methods when CutMix is the better choice. It would be better to provide the case when Mixup is alternatively the better choice to further support that the proposed methods combine the advantages of both these two methods.

2. The captions of figures could be more detailed.

**Strengths And Weaknesses:**

Strengths:

1. Extensive analysis to extend previous theoretical understanding of Mixup to most of the Mixup-based methods.
2. Empirical results to support their theoretical findings.
3. The overall flow is good, every time I have a question I see the answer.

Weakness:

1. The proposed new methods appear to have marginal improvements over current results, although the current best one - Stochastic Mixup & CutMix also supports their claims.
2. Some of the parts could be written in a better manner,  e.g., the explanation in Fig. 4.

---

> ### Author Response · Authors · 2022-08-02
> **Response to Reviewer ynng (1)**
>
>
> We thank Reviewer ynng for the thoughtful reviews. We are happy to hear that the reviewer found that (1) our analysis is extensive and our results extend previous theoretical understanding of Mixup to an arbitrary MSDA method, (2) our empirical studies are aligned with our theoretical findings, and (3) the overall writing is good. We address all concerns and questions raised by the reviewer  and revise the manuscript accordingly.
>
>
> ### Q1. Where is Theorem 4?
> Thank you for correcting our mistakes. We correct the numbering of Theorems in the revised version of the Appendix.
>
>
> ### Q2. The meaning of Fig 4 including x-axis, y-axis and color bar. & W2. Explanation of Fig 4 & L2. Captions
> Equation (8) shows that the regularization term $a_{ij}$ directly affects to the pixel gradients $|\partial_i f_\theta(x_k) \partial_{j} f_\theta(x_k)|$ in our approximated loss function. The purpose of Figure 4 is to show how the pixel gradients are actually regularized after training. To show that, we investigate the amount of the regularized input gradients by $|\partial_v f_\theta(x) \partial_{v+p} f_\theta(x)|$ with respect to the pixel distance vector $p$ for trained models by different MSDA methods. Here, if our approximated loss function actually behaves as a regularization, then we can expect that the pixel gradients $|\partial_v f_\theta(x) \partial_{v+p} f_\theta(x)|$ is small when $a_{ij}$ is large for the given $p$.
>
> We first define the partial gradient product as follows:
> $$\text{PartialGradProd}(x,p) = \max_{v} |\partial_v f_\theta (x) \partial_{v+p} f_\theta (x)|$$
> Now, we visualize the pixel-wise maximum values of PartialGradProd(x, p) in Figure 4. We train different models $f_\theta$ on resized ImageNet (64 x 64) and measure the values on the validation dataset. The x-axis and y-axis of Figure 4 denote the pixel distance $p$ along each x- and y- axis, and the scale of the colorbar denotes the value of the maximum partial gradient product. In the figure, we can observe that CutMix reasonably regularizes effectively in the input gradients products when a pixel distance is small; these results aligned with our previous interpretation, CutMix behaves as a pixel-level regularizer where it gives stronger regularization (larger $a_{ij}$) to the closer pixels.
>
> We agree with the reviewer that Figure 4 can be re-written better. We changed the description of Figure 4 (L256-L272 in the Supplementary Material), the details of Figure 4 and its caption in the revised version of the paper.
>
>
>
> ### W1. The proposed methods show marginal improvements although the current best one – Stochastic Mixup & CutMix – also supports their claims. & L1. The paper only provides the results when CutMix is better than Mixup, no converse case.
> We first re-emphasize that the main goal of our paper is to understand various MSDAs from the optimization perspective and to provide a generalized loss function form. We also have shown that there is no one-fit-all solution for MSDA, resulting in proposing a hybrid version of popular MSDA methods, i.e., Mixup and CutMix. More specifically, the loss function for MSDA is determined by $a_{ij}(=E_M[(1-M_i)(1-M_j)])$, and we claim that the best $a_{ij}$ is problem and data dependent. In Table 1, we show when Mixup is better than CutMix and the converse case. Mixup performs better than CutMix when the longer distance pixels are relative (e.g., larger objects and smaller crop size). To address the reviewer’s concern, we additionally report HMix results in Table 1. The results support that our proposed method takes the advantages from both sides. This table is in the revised version of Appendix I.1.
>
> |                        |  Mixup | CutMix | $\Delta$ (CutMix - Mixup) |     HMix ($r$=0.5)     |    HMix ($r$=0.75)    |
> |------------------------|:------:|:------:|:-------------------------:|:----------------------:|:---------------------:|
> | Scenario 1: Large crop |  58.3  | **64.4** |            +6.1           |  **61.3 (+3.0 vs. Mixup)** | **63.7 (+5.4 vs. Mixup)** |
> | Scenario 2: Small crop | **67.7** |  67.0  |            -0.7           | **67.6 (+0.6 vs. CutMix)** | **67.2 (+0.2 vs. CutMix)** |

---

> > ### Comment · Reviewer_ynng · 2022-08-04
> > **Thanks for the response**
> >
> > Thank you for preparing the rebuttal.
> >
> > The overall results look pretty good to me know. I'll raise my score to 7.
> >
> > One minor concern:
> >
> > 1. FGSM is quite an old attack, and I suggest using AutoAttack[1] instead.
> >
> > [1] Croce, F., & Hein, M. (2020, November). Reliable evaluation of adversarial robustness with an ensemble of diverse parameter-free attacks. In International conference on machine learning (pp. 2206-2216). PMLR.

---

> ### Author Response · Authors · 2022-08-02
> **Response to Reviewer ynng (2)**
>
>
> We additionally report robustness benchmarks, where CutMix and Mixup behave in a significantly different way; CutMix shows strong occlusion robustness (e.g., when an image is randomly occluded) and Mixup shows strong corruption robustness (e.g., when a Gaussian noise is added to an image [R1]) as shown by Chun et al [R2]. The following table shows the results on ImageNet standard accuracy, ImageNet FGSM accuracy (adversarial attack), ImageNet-occ accuray (center occluded images following Yun et al. [R3]), and ImageNet-C accuracy (Corrupted images by 15 corruptions proposed by Hendrycks et al. [R1]). This table is in the revised version of Appendix I.2.
>
> | Augmentation Method |   ImageNet-1K   |       ImageNet-occ      |       ImageNet-C       |           FGSM          |
> |---------------------|:---------------:|:-----------------------:|:----------------------:|:-----------------------:|
> | Vanilla (no MDSA)   |  75.68 (+0.00)  |      55.26 (+0.00)      |      42.57 (+0.00)     |       8.55 (+0.00)      |
> | Mixup               |  77.78 (+2.10)  |      60.34 (+5.08)      | **51.73 (+9.16)** |      27.78 (+19.23)     |
> | CutMix              |  78.04 (+2.36)  | **71.51 (+16.25)** |      44.18 (+1.55)     |      33.63 (+25.08)     |
> | HMix                |  **78.38 (+2.70)**  | **71.13 (+15.87)** | **46.37 (+3.80)** | **34.98 (+26.44)** |
> | GMix                | **78.13 (+2.45)** |      62.76 (+7.50)      |      45.97 (+3.40)     |      31.02 (+21.47)     |
>
>
>
> - In the case of the image dataset being corrupted by noises or blurs (ImageNet-C), we noticed that Mixup performs better than CutMix. This phenomenon can be explained by our theoretical results too. The ImageNet-C style corruption is globally applied to the image regardless of the content of the original image. In this case, the longer-distance relationships are significantly damaged and useless to distinguish the object, hence the shorter-distance relationships are more important. As HMix and GMix less weigh shorter-distance relationships than Mixup (but more weight than Mixup), we can observe that HMix and GMix ImageNet-C performances are better than CutMix, but worse than Mixup.
> - In the case of the image dataset having occlusions, we noticed that CutMix performs better than Mixup. In this case, the occluded areas have no information to distinguish objects, but only local areas are informative. Hence, it is important to capture shorter-relationship rather than global-relationship. As Scenario 1 in page 8, we can expect that CutMix is better than Mixup in this case. Not surprisingly, HMix and GMix are located in between CutMix and Mixup.
>
>
> [R1] Hendrycks, Dan, and Thomas Dietterich. "Benchmarking neural network robustness to common corruptions and perturbations." ICLR (2019).
>
> [R2] Chun, Sanghyuk, et al. "An empirical evaluation on robustness and uncertainty of regularization methods." ICML Workshop (2020).
>
> [R3] Yun, Sangdoo, et al. "Cutmix: Regularization strategy to train strong classifiers with localizable features." ICCV (2019).

---

### Author Response · Authors · 2022-08-02
**Common Comments**

Dear Reviewers,

We deeply appreciate your reviews. Thank you for having time to read our manuscript and giving good advice.

Firstly, we want to emphasize the novelty and contribution of our work. Our paper provides a unified analysis of MSDA, including Mixup and CutMix, and this is the first work that analyzed the different effects of MSDA with a theoretical lens to the best of our knowledge. We provided the hypothesis that Mixup or CutMix will perform well with our theorems. We also provided new methods and experiments to validate our theorem.

Second, we revised our paper threefold:

(1) We added some experiments to support our theorem. Moreover, we added the robustness benchmark (e.g., Imagenet-C, ImageNet occlusion, adversarial attack) to provide more abundant examples of when Mixup or CutMix perform well. Lastly, the validation for the second order Taylor expansion is also provided with experiments.

(2) We changed the description of Figure 4. In the previous version, the explanation of Figure 4 was insufficient, as mentioned in the review. We make the description of Figure 4 clear.

(3) We changed the minor errata in the paper.
The revised paper is attached in the supplementary material.

If further inquiries come up within the rebuttal period, we would be pleased to talk with the reviewer again!

Best,

Authors

---

### Meta-Review · Area_Chair_L6C7 · 2022-08-31

**Recommendation:** Accept
**Confidence:** Certain

**Metareview:**

This work proposes a theoretical analysis and unified specification for mixed sample data augmentation methods. The reviewers praise the extensive theoretical analysis as well as the strong empirical results in the paper. The authors and reviewers engaged in substantial discussion, which led multiple reviewers to revise their assessment of the paper upwards. I can therefore recommend accepting this paper.

**Award:**

No

---

### Decision · Program_Chairs · 2022-09-14

Accept